# Identifying Crop Growth Stages from Solar-Induced Chlorophyll Fluorescence Data in Maize and Winter Wheat from Ground and Satellite Measurements

Yuqing Hou [1,2], Yunfei Wu [2,3], Linsheng Wu [2,3], Lei Pei [2,3], Zhaoying Zhang [1,2], Dawei Ding [4], Guangshuai Wang [4], Zhongyang Li [4] and Yongguang Zhang [1,2,3,*]

1. International Institute for Earth System Sciences, Nanjing University, Nanjing 210023, China; yqhou@smail.nju.edu.cn (Y.H.); zhaoying_zhang@nju.edu.cn (Z.Z.)
2. Jiangsu Provincial Key Laboratory of Geographic Information Science and Technology, Key Laboratory for Land Satellite Remote Sensing Applications of Ministry of Natural Resources, School of Geography and Ocean Science, Nanjing University, Nanjing 210023, China; dz1827003@smail.nju.edu.cn (Y.W.); wulinsheng@smail.nju.edu.cn (L.W.); 502022270077@smail.nju.edu.cn (L.P.)
3. Huangshan Park Ecosystem Observation and Research Station, Ministry of Education, Huangshan 245899, China
4. National Agro-Ecological Observation and Research Station of Shangqiu, Institute of Farmland Irrigation of CAAS, Shangqiu 476002, China; dingdawei@caas.cn (D.D.); wangguangshuai@caas.cn (G.W.); lizhongyang1980@163.com (Z.L.)
* Correspondence: yongguang_zhang@nju.edu.cn; Tel.: +86-25-89681569

**Abstract:** Crop growth stages are integral components of plant phenology and are of significant ecological and agricultural importance. While the use of remote sensing methods for phenology identification in cropland ecosystems has been extensively explored in previous studies, the focus has often been on land surface phenology, primarily related to the start and end of the growing season. In contrast, the monitoring of crop growth within an agronomic framework has been limited, particularly in the context of recently developed solar-induced chlorophyll fluorescence (SIF) data. Additionally, some critical growth stages have not received adequate attention or evaluation. This study aims to assess the utility of SIF data, collected from both ground and satellite measurements, for identifying critical crop growth stages within the realm of remote sensing phenological estimation. A comparative analysis was conducted using enhanced vegetation index (EVI) data at the Shangqiu site in the North China Plain from 2018 to 2022. Both SIF and EVI time-series data, obtained from ground and satellite sources, undergo a comprehensive phenological estimation framework encompassing pre-processing, modeling, and transition characterization. This approach involves reconciling time-series phenological patterns with crop growth stages, revealing the necessity of redefining the mapping relationship between these two fundamental concepts. After preprocessing the time-series data, the framework incorporates the phenological modeling process employing two double logistic models and a spline model for comparison. Additionally, it includes phenological transition characterization using four different methods. Consequently, each input dataset undergoes an assessment, resulting in 12 sets of estimations, which are compared to select the ideal estimation portfolio for identifying the growth stages of maize and winter wheat. Our findings highlight the efficacy of SIF data in accurately identifying the growth stages of maize and winter wheat, achieving remarkable results with an R-square exceeding 0.9 and an RMSE of less than 1 week for key growth stages (KGSs). Notably, SIF data demonstrate superior accuracy, robustness, and sensitivity to phenological events when compared to EVI data. This study establishes an estimation portfolio utilizing SIF data, involving the Gu model, a double logistic model, as the preferred phenological modelling method together with various compositing methods and transition characterization methods, suitable for most KGSs. These findings create opportunities for future research aimed at enhancing and standardizing crop growth stage identification using remote sensing data for a wide range of KGSs.

**Keywords:** crop growth stage; SIF; EVI; phenology extraction; time series

## 1. Introduction

The dynamics of periodic events in plants, known as plant phenology, reflect the responses of the terrestrial biosphere to climate change [1–4]. As the specific part of plant phenology, the agricultural crop growth stages function as more sensitive and integrated indicators of the agricultural ecosystem to environmental conditions [5,6]. In addition, as the physiological signals responding to the growing conditions, crop growth stages could determine the biophysical structure of the crop and its photosynthesis intensity, evapotranspiration rate and light use efficiency [7,8]. For example, drought stress during the silking stage of maize could cause a 3–8% loss to its yield, while, for soybeans during the setting pods stage, the reduced number of pods may be 20% [9,10]. That explains the reason why the yield loss of maize and soybean are particularly sensitive to the water condition during the silking stage and the setting pods stage, respectively [11,12]. For the same reason, these two growing stages have been also seen as the critical irrigation window of these two crops [12,13]. Hence, the agricultural crop growth stages could build a critical link between environmental conditions and the physiological growth of crops, and contribute to crop yield estimation, precise crop management and decision making.

The solar-induced chlorophyll fluorescence (SIF) is the signal emitted by the chlorophyll-a molecules in vegetation when they absorb photosynthetically active radiation during photosynthesis. This signal, which falls within the spectrum of 650 nm to 800 nm, is a valuable indicator of various plant phenophases [14]. As a by-product of photosynthesis, SIF has been proven to be strongly related with the gross primary production (GPP) of vegetation for different ecosystems [15–18]. In addition to its role in monitoring photosynthesis, SIF also provides valuable insights into the biophysical structure of plants [19–21]. Due to its advantages over traditional vegetation indices (VIs), SIF data has become increasingly popular for monitoring plant growth conditions [15,22,23]. Over the past decade, SIF data has been retrieved globally from various satellite missions, including GOSAT (Greenhouse gases Observing SATellite) [15,16,22], GOME-2 (The Global Ozone Monitoring Experiment-2) [24,25], SCIAMACHY [25,26], OCO-2 (Orbiting Carbon Observatory-2) [27], TROPOMI (TROPOspheric Monitoring Instrument) [28] and Tansat [29,30]. These missions have paved the way for regional-scale crop growth condition monitoring using SIF data.

Advancements in remote sensing technology have opened new avenues for understanding plant phenology, offering valuable insights into crop growth and its response to environmental factors [31–35]. Phenological estimation utilizing remote sensing time series data typically involves several key steps. Initially, the data undergo compositing and filtering to mitigate cloud contamination, viewing geometry issues, and atmospheric interference [36,37]. Subsequently, phenological models such as smoothing spline functions, double logistic functions, and asymmetric Gaussian functions are employed to fit the signal curve throughout the growing season [31,38,39]. Finally, phenological transition dates are extracted from the fitted phenological curve with various methods [40–42]. Despite the progress made in utilizing remote sensing data for the crop phenology estimation, several challenges persist. Firstly, there is a discrepancy between phenological transition dates, such as the start of the season, and crop growth stages like the silking stage of maize and heading stage of winter wheat, creating a temporal gap [5,42]. Secondly, many studies on crop phenology estimation are constrained within the broader framework of plant phenology, focusing primarily on the start, peak, and end of the growing season [43–45]. This limitation hinders the alignment of phenological estimations with actual crop growth stage observations in agronomy. Thirdly, while SIF data has been widely proven effective in tracking GPP and crop growth observation, its potential for identifying specific crop growth stages remains unexplored [45,46].

This study focuses on a specific agronomic observation site situated in the North China Plain, aiming to harness the potential of SIF data obtained from both ground-based measurements and satellite retrievals. With a specific emphasis on two key crops, summer maize and winter wheat, this research endeavors to achieve the following objectives: (1) Alignment of phenological characteristics: To establish a clear correspondence between

the time-series phenological characteristics derived from the remote sensing data and the actual growth stages observed on the ground; (2) SIF data for growth stage estimation: Assessing the capability of SIF data to effectively estimate critical crop growth stages within the framework of agronomy; (3) Comparison with vegetation indices: A comparative analysis to determine whether SIF data exhibits superior performance when compared to traditional VIs (EVIs) in accurately identifying and characterizing crop growth stages; (4) Exploration of acceptable portfolios: Exploring and identifying acceptable combinations of remote sensing time-series phenological estimation processes encompassing pre-processing, modeling, and transition characterization that prove effective in the precise identification of crop growth stages.

## 2. Materials and Methods

### 2.1. Study Site

The study site, located in Shangqiu, Henan Province, China (Figure 1), is a part of the ChinaSpec network (https://chinaspec.nju.edu.cn/(accessed on 6 December 2022)) managed by Nanjing University and Institute of Farmland Irrigation of the Chinese Academy of Agricultural Sciences (CN-SQ, National Agro-Ecological Observation and Research Station of Shangqiu: 34.5203°N, 115.5894°E) [46–48]. This region in the North China Plain is renowned for its annual crop rotation, mainly involving winter wheat (*Triticum aestivum*) and summer maize (*Zea mays*). The climate is characterized by a warm and semi-humid continental monsoon, with an average annual precipitation of 706 mm and a mean temperature of 13.9 °C. Winter wheat is planted in mid-October, followed by a winter season with snowfall. The regreening stage of winter wheat typically begins in early March, followed by the harvest of winter wheat in June. Subsequently, summer maize is planted and harvested in late September to early October, starting a new crop rotation cycle.

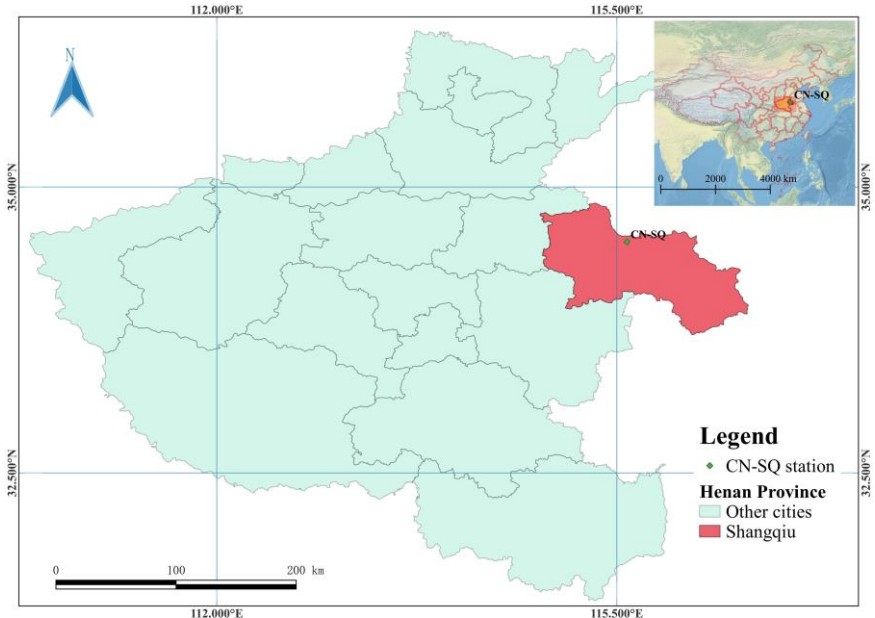

**Figure 1.** The map of the study site (CN-SQ), marked with the green diamond in this figure. Shangqiu has been marked red in the map of Henan Province, with other cites painted royal blue. The reference map is at the up-right corner, indicating the location of the Henan Province is at the heart of North China Plain as the center of one main grain-producing area in China.

### 2.2. Observed Crop Growth Stages

As detailed in Section 2.1, the observation of crop growth stages at the CN-SQ site is conducted in collaboration with the National Agro-Ecological Observation and Research Station of Shangqiu, Institute of Farmland Irrigation of the Chinese Academy of Agricul-

tural Sciences. The dedicated team from this institute undertakes comprehensive crop growth stage observations, encompassing not only the experimental farmland within the study site but also the broader agricultural landscape in the vicinity. Ground-based spectrum measurements, or in situ measurements, were conducted within the experimental farmland of the study site. These observed growth stages within the station served as reference data for estimating crop growth stages at the site level (site-OBs). Additionally, considering the broader coverage of satellite data over a larger agricultural landscape surrounding the study site, the average of the observed growth stages from both within the site and the broader agricultural area in proximity (area-OBs) were utilized as reference data for satellite-based estimations. The observations are meticulously recorded to document various agronomic growth transitions and milestones. The recorded crop growth stages and their respective definitions are presented in Table 1, sourced from ground observers' expertise and aligned with the Integrated Crop Management Handbook and Best Management Practices for Wheat Production [49,50].

**Table 1.** Ground-observed crop growing stages.

| Crop | Growth Stages | Definition |
|------|---------------|------------|
| Winter wheat | Regreen (RG) | The plant turns green again after the winter. |
| | **Jointing (JT)** | From 1st node detectable to last leaf visible. |
| | **Heading (HD)** | Head is fully exposed to frost, hail and pests. Plant attains final height. |
| | **Milk (MK)** | Starch and protein content determination starts, or namely 'grain filling'. |
| | Ripening (RP) | Kernel hard, difficult to divide by thumbnail. The plant is completely yellow. |
| | Harvest (HV) | The plant has been harvested. |
| Summer maize | Sowing (SOW) | Seeds have been planted in the soil. |
| | Emergence (VE) | Shoot (coleoptile) has emerged from the soil. |
| | **5th leaf (V5)** | The 5th leaf collars present. |
| | **Jointing (JT)** | Between V6 and V9, the first stem of maize grows to the height approximately 2 cm. |
| | **Tasseling** | Lowest branch of the tassel is visible. |
| | **Silking** | One or more silks extends outside of husk leaves. |
| | **Milk (MK)** | Kernels filled with 'milky' fluid, or namely 'grain filling'. |
| | Maturity (MT) | Kernels at maximum dry matter accumulation; a 'black layer' will form at kernel base (2–3 days after physiological maturity). |
| | Harvest (HV) | The plant has been harvested. |

Note: The bolded growth stages in this table are key growth stages (KGSs).

For summer maize, the tasseling and silking growth stages occur in close succession, typically within a 1- to 3-day window. Given the potential margin of error in observation, this study simplifies this by treating these stages as one, referred to as tasseling and silking (T&S). Similarly, the maturity and harvest stages, often temporally intertwined, are combined into a single stage named maturity and harvest (M&H).

Moreover, not all growth stages have an equal impact on crop yield. For instance, in maize cultivation, the silking stage is highly sensitive to water stress, necessitating meticulous irrigation management [51,52]. Additionally, the V5 and jointing stages, when the stem of maize grows fast described in Table 1, demand increased attention due to maize's heightened need for nitrogen fertilizer and water during these phases [53–57]. Similarly, both wheat and maize assign critical importance to the milk stage [49,58]. During this phase, kernels undergo crucial filling, exerting a substantial influence on the overall yield potential. Recognizing the pivotal nature of these growth stages, this study designates them as key growth stages (KGSs), which have been bolded in Table 1.

*2.3. Ground-Based Spectrum Measurements*

Seasonal canopy reflectance and SIF observations were conducted using an automated ground-based continuous observation system known as AS-SpecFOM, (Agri-SIF Envi-

ronmental Technology Co., Ltd., in Nanjing, China). This system bears a resemblance to FluoSpec2 and comprises two essential components [59]. The core of the system features a QEPRO spectrometer (Ocean Optics, Dunedin, FL, USA) with a spectral range spanning from 730 to 785 nm, a full-width half-maximum (FWHM) of 0.17 nm, and an impressive signal-to-noise ratio (SNR) of approximately 1000. This QEPRO spectrometer is primarily utilized for the retrieval of SIF data. In conjunction with the QEPRO, the system incorporates an HR2000+ spectrometer (Ocean Optics, Orlando, FL, USA) with a broader spectral range, covering wavelengths from 350 to 1000 nm, an FWHM of 1.1 nm. The HR2000+ is dedicated to capturing canopy reflectance measurements and vegetation indices (VIs). Specifically, the enhanced vegetation index (EVI) was computed using Equation (1) [60]. In Equation (1), $\rho_{NIR}$, $\rho_{red}$ and $\rho_{blue}$ are reflectance at near-infrared, red and blue bands after the atmospheric correction. The entire system boasts a 25° field of view (FOV) and is positioned at an approximate height of 10 m above the canopy.

$$EVI = 2.5 \times \frac{\rho_{NIR} - \rho_{red}}{\rho_{NIR} + 6 \times \rho_{red} - 7.5 \times \rho_{blue} + 1} \tag{1}$$

Table 2 provides an overview of the in situ SIF and EVI data collected for winter wheat and maize during specific years. In the case of winter wheat, data were systematically gathered in 2019, 2021 and 2022. The absence of data for winter wheat in 2018 and 2020 was attributed to the construction and the relocation of the observation system, respectively. For maize, in situ SIF measurements were consistently acquired from 2018 to 2022, while in situ EVI data were obtained from 2019 to 2022. The lack of in situ EVI data in 2018 resulted from data loss. To mitigate the impact of cloud cover on in situ SIF measurements and the photosynthetic activity of crops, measurements conducted during periods with a clearness index (defined as actual shortwave radiation divided by top-of-atmosphere shortwave radiation) lower than 0.5 were identified as observations made under cloudy conditions and were consequently excluded from the dataset [61]. To identify the best observation time period for each data, the in situ measurements were categorized into three groups: 'Morning', 'Afternoon' and 'Whole-day', each representing a 1-day resolution with the mean values recorded within the respective time periods. Specifically, 'Morning' corresponds to the time frame between 7:00 AM and 12:00 PM, while 'Afternoon' encompasses the period from 12:00 PM to 5:30 PM.

**Table 2.** Data collection in this study.

| | Data | Time Cover | Temporal Resolution | Spatial Resolution |
|---|---|---|---|---|
| Ground-based data | Ground-measured SIF (In situ SIF) | Maize: 2018–2022 Wheat: 2019, 2021, 2022 | 0.5 h | - |
| | Ground-measured EVI (In situ EVI) | Maize: 2019–2022 Wheat: 2019, 2021, 2022 | 0.5 h | - |
| Satellite data | TROPOMI SIF | 2018–2022 | 1 day | 0.05° |
| | MODIS EVI | 2018–2022 | 8/16 days | 0.05° |

*2.4. Satellite Data*

2.4.1. TROPOMI SIF

TROPOMI, carried aboard the Sentinel-5 Precursor satellite, commenced its mission on 13 October 2017, with co-funding from the European Space Agency (ESA) and the Netherlands. This advanced instrument provides a wide swath of approximately 2600 km, delivering high-resolution spatial measurements. Prior to 6 August 2019, TROPOMI boasted a spatial resolution of 3.5 km × 7 km, which transitioned to 5.6 km × 3.5 km after the specified date. TROPOMI enables the retrieval of SIF data within spectral ranges of 735–758 nm and 743–758 nm. Then, all of its products are normalized to 740 nm using a reference fluorescence spectrum [62].

As detailed in Table 2, the ungridded TROPOMI SIF740 product from 2018 to 2022 was sourced from ftp://fluo.gps.caltech.edu/data/tropomi/ungridded/ (accessed on 5 September 2023) (for additional details, refer to: https://doi.org/10.22002/D1.1347 (accessed on 5 September 2023)) for utilization in this study. To enhance data quality, measurements with viewing zenith angles (VZAs) exceeding 60° were excluded due to the substantial uncertainties at the swath edges. Additionally, data influenced by dark/bright scenes and cloud cover were systematically removed [28]. This study utilizes daily average SIF data, adjusting the instantaneous SIF records within a 20 km radius from Shangqiu Station using a daily-length correction factor (denoted as 'dcsif' in the product). Furthermore, instantaneous SIF records featuring a cloud fraction (denoted as 'cf' in the product) exceeding 0.2, identified as cloud-contaminated records, were omitted from the dataset.

### 2.4.2. MODIS EVI

The Moderate Resolution Imaging Spectroradiometer (MODIS) aboard the Terra (originally known as EOS AM-1) and Aqua (originally known as EOS PM-1) satellites have been widely employed in plant growth observation across extensive geographical regions [45,63–67]. MODIS provides spatial resolutions of 250 or 500 m and a temporal cadence of 1–2 days, facilitating global-scale dynamic plant growth monitoring.

As presented in Table 2, this study obtained MODIS-derived EVI from the MOD13C1 (Terra) and the MYD13C1 (Aqua) Version 6 product spanning from 2018 to 2022. Combining these two daily 16-day composite datasets, a corresponding EVI time series was generated at an 8-day resolution. The spatial resolution of the EVI data are set at 0.05°, aligning with the spatial resolution of TROPOMI SIF data for a consistent resolution-level comparison.

### 2.5. Methods

This study's primary objective is to assess the performance of SIF data within the context of remote sensing phenological monitoring, aligning it with ground-based crop growth observations. The methodology encompasses three core components: time-series phenological pre-processing (Section 2.5.1), time-series phenological modeling (Section 2.5.2), and physiological transaction characterization (Section 2.5.3) [6,66]. These latter two sections leverage the R module Phenopix to enhance efficiency [68].

The phenological framework employed here features a meticulously designed set of methodologies tailored for agricultural phenological monitoring. These methodologies include seasonality filtering to mitigate the impact of weeds and cover crops, double logistic-based phenological models to capture rapid changes in crop growth and development, and a range of phenological characterization methods to ascertain a comprehensive spectrum of crop growth stages.

### 2.5.1. Time-Series Phenological Pre-Processing

The time-series data collected for this study are susceptible to various sources of disturbance, including diverse radiance conditions, background interference, instrumental noise, and atmospheric factors. Such abnormalities or implausible observations can obfuscate or even disrupt the underlying phenological patterns of the target crops. Consequently, the primary objective of the pre-processing phase is to mitigate the impact of the atmospheric, background and systematic noise while smoothing the time-series data, thus preparing it for subsequent phenological modeling. This pre-processing stage comprises two key components:

1.  Abnormal measurements elimination: To address abnormal measurements, a moving window abnormal elimination method was employed. This approach identifies measurements deviating by more than three times the standard deviation from the average value within an 11-day moving window (slightly longer than the interval of adjacent observed crop growth stages) and subsequently excludes them from the dataset [69].

2. Time series smoothing: SIF and EVI time series underwent smoothing using a three-time Savitzky–Golay algorithm with an 11-day moving window [70]. This algorithm serves to attenuate off-season phenological signals while fitting a smoothing curve to the time-series observations.

As illustrated in Figure 2, this pre-processing phase effectively transforms the raw data points into the smoothed dashed lines.

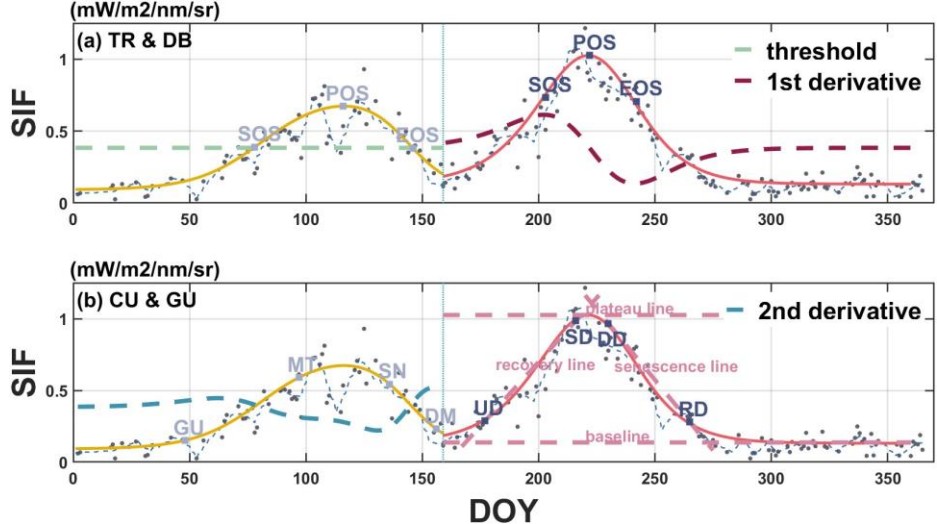

**Figure 2.** The variations observed in four distinct phenological characterization methods employed in this study. The dots represent the original crop measurements, while the dashed lines illustrate the fitted lines following the pre-processing step. Solid lines represent the phenological modeled curves, with winter wheat in yellow and maize in red. Vertical dashed lines in light blue indicate the auto-recognized rotation points between winter wheat and maize. (**a**) Displays transitions extracted using the threshold-based (TR) method for winter wheat (yellow) and the derivative-based (DB) method for maize (red). (**b**) Showcases transitions extracted using the curvature-based (CU) method for winter wheat (yellow) and the Gu (GU) method for maize (red).

The second aspect of this pre-processing section involves categorizing and compositing the datasets into intervals with different temporal resolutions, facilitating subsequent processing and evaluation. Initially, based on empirical knowledge of crop rotation detailed in Section 2.1, the minimum value between the day of the year (DOY) 100 and 200 serves as the rotation point. This division enables the segmentation of the smoothed time-series curve into two distinct growing seasons: winter wheat and maize. Notably, this step is exclusively applied to satellite data, because the in situ measurements were already classified. As depicted in Figure 2, the dashed lines in light blue indicate the rotation points derived through this procedure. Subsequently, the smoothed time-series curve is composited into data intervals spanning 3 days, 5 days, 7 days and 15 days, employing both maximum value composite (MVC) and average value composite (AVC) techniques at the midpoint of each interval [71]. Applying this step to MODIS EVI data is worth nothing, as the temporal resolution of the MODIS EVI product used in this study was 16 days and 8 days (when combined), and it had already undergone compositing.

### 2.5.2. Time-Series Phenological Modelling

Following the phenological pre-processing of the time-series curve, the next step involves modeling the seasonal smoothed phenological curve to elucidate its developmental trajectory and capture rapid changes during the growing season. This modeling phase is depicted in Figure 2, transitioning from the smoothed dashed lines to the solid yellow or red lines. For this modeling task, the study opted for double logistic functions, a type of curve-fitting-based phenological model, owing to their widespread utilization

in phenological monitoring and superior performance compared to other smoothing algorithms [31]. Furthermore, double logistic functions exhibit better modeling performance for relatively short and rapid-growing seasons, which are common in agricultural crops, when contrasted with alternative algorithms like Fourier analysis and asymmetric Gaussian functions [72]. In this study, two variants of double logistic functions, namely the Beck and Gu models, were employed, alongside the smoothing spline function as a simpler alternative for comparison.

The Beck model, which establishes a minimal baseline for the growing-season curve, has been shown to mitigate the influence of spurious measurements during the fitting process [72]. Beck model characterizes the temporal variations of the growing season using six parameters:

$$f(t) = a_{base} + (a_{max} - a_{base}) \times \left( \frac{1}{1 + e^{(-m_1 \times (t - m_2))}} + \frac{1}{1 + e^{(n_1 \times (t - n_2))}} - 1 \right) \quad (2)$$

In the equation, $t$ represents DOY, and $f(t)$ represents the fitted value at $t$. Additionally, $a_{base}$ signifies the minimum baseline or off-season value, while $a_{max}$ represents the maximum value during the growing season. Double logistic functions posit that two piecewise logistic functions of time, one for the upward and the other for the downward trend, can effectively depict the phenological development of the monitored plant [31]. Here, $m_2$ and $n_2$ denote the inflection points corresponding to the onset and conclusion of the ascending and descending portions of the pre-processed curve, while $m_1$ and $n_1$ denote the rates of increase and decrease at $m_2$ and $n_2$, respectively.

The Gu model, in contrast, offers greater generality compared to the Beck model, encompassing additional parameters for accommodating diverse phenological curve fitting requirements [73]. These supplementary parameters enhance the model's robustness and adaptability in tracking phenological trajectories. However, the broader set of parameters may pose challenges in phenological modeling when working with lower-quality original data. The function for the Gu model is as follows:

$$f(t) = a_0 + \frac{a_1}{\left(1 + e^{-\frac{t - m_2}{m_1}}\right)^{m_4}} - \frac{a_2}{\left(1 + e^{-\frac{t - n_2}{n_1}}\right)^{n_4}} \quad (3)$$

In the provided equation, $t$ represents the DOY, and $f(t)$ signifies the fitted value at $t$. The parameters $a_0$, $a_1$, $a_2$, $m_1$, $m_2$, $m_4$, $n_1$, $n_2$ and $n_4$ are the empirical parameters specific to the Gu model. These parameters are going to be literately set to fit the smoothed time-series data. This procedure was conducted using the R model Phenopix mentioned in Section 2.5 [68].

The spline model employed In this study is the smoothing spline function. This function is capable of fitting the seasonal phenological curve while also effectively removing outliers and off-season signals. It achieves this by fitting piecewise polynomials to temporal segments of the pre-processed phenological curve and subsequently joining these polynomials into one continuous curve. The spline model not only models the seasonal trajectories but also demands the continuity of both the modeled curve and its derivative [38]. As a data-driven function, the spline model does not impose constraints on the shape of the modeled curve, unlike double logistic functions. This characteristic makes it an ideal alternative to the two double logistic functions previously introduced, especially when dealing with data of lower quality, as it can effectively capture the complex patterns exhibited by crops.

### 2.5.3. Phenological Transition Characterization

With the phenological time-series curve smoothed and modelled as described in the previous sections, the focus now shifts to extracting critical transition points from the phenological curve for estimating crop physiological growth stages such as the jointing, tasseling, and silking stages. This study employs four widely recognized phenological

characterization methods: threshold-based (TR), derivative-based (DB), curvature-based (CU) and Gu-based (GU), to identify these transition points. The transition points and corresponding lines for these methods are illustrated in Figure 2.

The threshold-based (TR) method estimates three transition points in the phenological development curve by defining specific thresholds [41]. The thresholds can be defined as absolute values or relative values of the growing-season maximum value (amplitude of the curve) [74]. Relative studies have examined the performance of different thresholds settings in phenological monitoring [5,74–76]. Because the SIF and EVI data adopted in this study range differently in growing-season phenological observation, with EVI data ranges from 0 to approximately 1 and the value of SIF data could exceed 1 or even 1.2, an absolute value as the threshold for both SIF and EVI data is not appropriate for further comparation. So, this study chose the 50% of the amplitude of the growing-season phenological curve as the threshold. Start of season (SOS) and end of season (EOS) are defined as the time when the threshold being reached at the upward and downward direction of the phenological curve, respectively, while peak of season (POS) is at the peak of the curve.

The derivative-based (DB) method identifies the SOS, POS and EOS by examining local extremes in the first derivative of the phenological time-series curve. Specifically, as it is shown in Figure 2, the SOS and EOS of the DB method correspond to the absolute maximum and minimum points, respectively, of the first derivative curve of the model-fitted curve. And POS of DB method is defined as the zero point of the first derivative curve between the SOS and EOS. From the definition, it is obvious that the POS in the DB method aligns with the POS in the TR method.

The curvature-based (CU) method characterizes crop phenophases based on local extremes in the rate of change (the second derivative) of the curvature of the phenological curve [31,33]. As it is presented in Figure 2, this method extracts four transition points: greenup (GU), maturity (MT), senescence (SN) and dormancy (DM). GU and MT correspond to the two local maxima points in the second derivative curve of the first-half growing season, while SN and DM are defined as the two local minima points in the second derivative curve of the second-half growing season. If the maxima or minima cannot be reached, the CU method will keep on tracking the extreme points until reaching the boundary of the time range. In this study, the rotation points were recognized to divide the phenological observation of the whole year into two growing seasons of winter wheat and maize, which could cause a rapid change at the rotation points as it is illustrated in Figure 2. For winter wheat, some mis-recognitions of DM may occur due to the rapid change at the rotation points dividing the phenological observation of the entire year into two growing seasons.

The Gu-based (GU) method is the most complex of the four, capturing four transition points: upturn (UD), stabilization (SD), downturn (DD), and recession (RD). It achieves this through a combination of local extremes in the first derivative curve and the boundary lines [73]. The boundary lines consist of a baseline and a plateau line, marking the minimum and maximum of the phenological curve, respectively. Additionally, this method models the trajectory of the growing-season phenological curve with a recovery line in the first-half growing season and a senescence line in the second-half. The recovery line is defined as the line going through the maximum point of the 1st derivative curve with the maximum value of the first derivative curve as the slope. Correspondingly, the senescence line is the line going through the minimum point of the first derivative curve with the minimum value of the first derivative curve as the slope. As it is shown in Figure 2, UD and SD are determined when the recovery line intersects the baseline and plateau line, respectively, while DD and RD are estimated when the senescence line intersects the plateau line and baseline.

In summary, the TR method uses a threshold at 50% of the amplitude, while both the DB and GU methods rely on the first derivative curve. The CU method, on the other hand, is based on the second derivative curve. These methods are independent of the absolute values of the observed data and focus on the trend of the phenological curve. Variations in

the SIF and EVI values do not affect the results, making this approach applicable to other datasets. It is important to note that transition points obtained by these four phenological transition characterization methods differ due to variations in empirical thresholds and properties. As the relationship between these transition points in the phenological curve and crop growth stages remains unrevealed at this time, the diversity of these points allows for the selection of the most suitable method or a combination of methods for different growth stage identifications.

*2.6. Accuracy Assessment*

To assess the performance of estimations employing various data and phenological identification portfolios, several common statistical measures of agreement were adopted in this study. These measures include the coefficient of determination ($R^2$), root-mean-square-error (RMSE), correlation, and standard deviation. $R^2$, RMSE and correlation assess the agreement between the estimated growth stages and the observed or referenced growth stages, while standard deviation provides information about the precision of the estimations.

For the comparison between different data sources, data-measured time periods, phenological modeling methods and transition characterization methods, the estimations from specific categories were sequentially evaluated using the aforementioned indices. As outlined in Section 2.2, estimations with in situ data were assessed based on site observations within CN-SQ station, while those with satellite data were evaluated using area observations, encompassing the adjacent agricultural landscape. The accuracy of estimation for each crop growth stage, especially for key growth stages (KGS), was evaluated based on the reconciled time-series phenological characteristics with the appropriate method.

The accuracy assessment results were used to identify the better dataset, data-measured time period, phenological modeling method or transition characterization method with superior accuracy. Ideally, these components could contribute to forming a phenological estimation portfolio suitable for various scenarios. Finally, the estimations made by the portfolio were evaluated through the $R^2$ and RMSE of linear regression with the corresponding observations, depending on the data utilized in the portfolio. It is important to note that the $R^2$ and RMSE for the portfolio are based on the total accuracy of several growth stages, while the accuracy assessment for specific components is based on individual growth stages.

## 3. Results

*3.1. Reconciliation of Time-Series Phenological Characteristics with Crop Growth Stages*

As discussed earlier, there exists a gap between the transitions identified in the smoothed and model-fitted phenological curve and the actual crop physiological growth stages [66]. To bridge this gap, a reconciliation step is imperative to establish the mapping relationship between these two datasets before proceeding with accuracy assessment and further comparisons. In this study, the range of each estimated transition dataset was visualized and compared with the distribution of ground-based observations of crop growth stages (refer to Figures 3 and 4). If the majority of ground-based observations fall in the range of a specific transition date (e.g., the SOS in the TR method), it is highly likely that this crop growth stage aligns with the identified transition date and vice versa. This comparative analysis allows us to establish potential mapping relationships between transition dates and crop growth stages, which will be subject to further assessment.

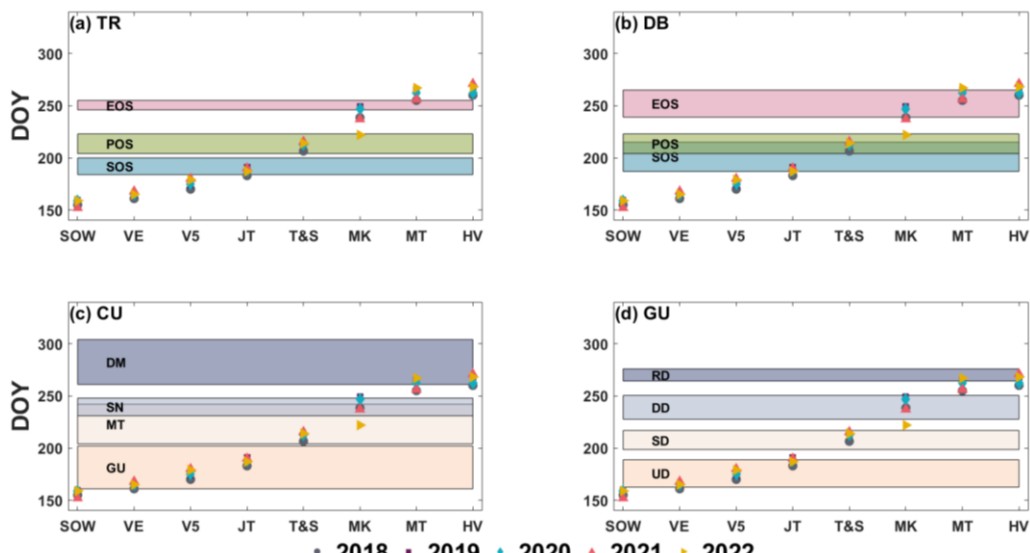

**Figure 3.** Comparation between ground observation of maize growth stages from 2018 to 2022 and transition dates estimated by the phenological identification framework (spline model) with TROPOMI SIF. Each dot on the graph corresponds to observations from a different year. The distribution of transition dates derived through various methods—threshold-based (TR), derivative-based (DB), curvature-based (CU), and Gu-based (GU)—is visualized using distinct colors in panels (**a**–**d**).

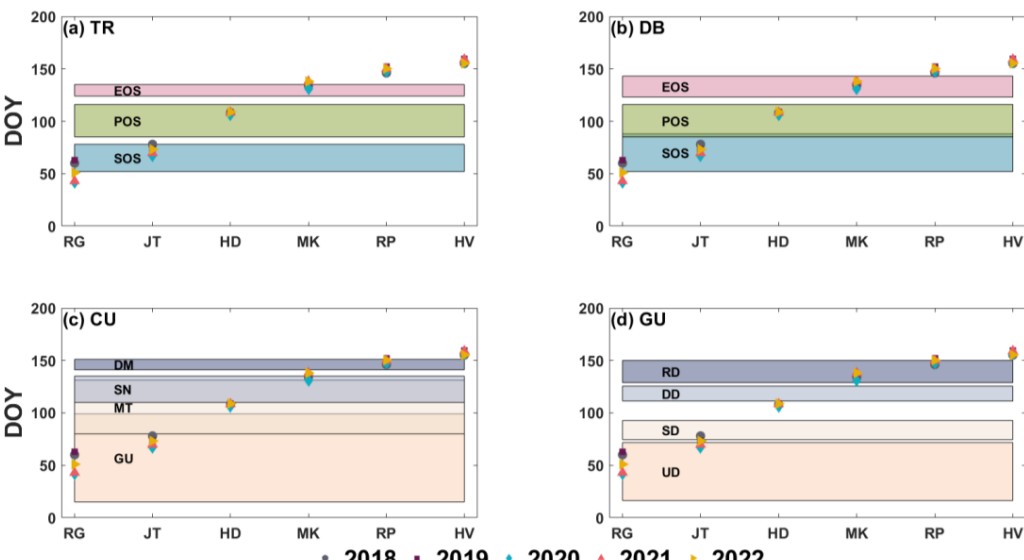

**Figure 4.** Comparison between ground-observation of winter wheat growth stages from 2018 to 2022 and transition dates estimated by the phenological identification framework (spline model) with MODIS EVI. Each dot on the graph corresponds to observations from a different year. The distribution of transition dates derived through various methods—threshold-based (TR), derivative-based (DB), curvature-based (CU), and Gu-based (GU)—is visualized using distinct colors in panels (**a**–**d**).

In Figures 3 and 4, the distinction between precision and accuracy in estimation is illustrated. Taking Figure 3a as an example, the SOS, POS and EOS extracted by the TR method span from approximately DOY 185 to 200, 205 to 225 and 245 to 255, respectively. Each set of these transition dates exhibits a high degree of precision, indicating a very narrow time span. However, notably, the first three growth stages (SOW, VE and V5) are not included in the range of any of these transition dates, suggesting that none of these three transition dates are suitable for identifying these initial growth stages. In contrast,

the JT and T&S stages are entirely encompassed within the value range of SOS and POS, respectively, indicating that the SOS of the TR method may accurately correspond to JT, and the POS of the TR method could be used to accurately identify T&S. For the MK, MT and HV stages, which are more commonly distributed around the time span of EOS, suggesting that the EOS of the TR method may not correspond accurately to any of these three crop stages. Nevertheless, EOS remains the most suitable transition for estimating MK, MT and HV among the SOS, POS and EOS of the TR method, as these three growth stages are closest to its time span. Notably, the milk (MK) stage in 2022 occurred much earlier than in other years, possibly due to waterlogging in that particular year.

The transition dates extracted by various phenological characterization methods may exhibit differences in precision and variation, as depicted in Figure 3a. In Figure 4a, the transition dates of the TR method, Figure 4b for DB method and Figures 3d and 4d for the GU method demonstrate good precisions, with distinct and narrow time spans. However, the estimations from the DB method in Figure 3b and CU method in Figures 3c and 4c exhibit some overlapping sets of transition dates. For instance, in Figure 3b, POS and SOS of DB method overlap to the extent that they nearly exclude the JT stage from the SOS range and include the T&S stage in the POS range. Nevertheless, due to the empirical and physiological order between these two phenophases, where T&S must occur after JT, it is reasonable to consider SOS and POS as better suited for JT and T&S, respectively. While this distribution of disorder does not necessarily imply low precision in the estimated results, it can introduce uncertainty when reconciling time-series phenological characteristics with crop growth stages.

Moreover, the lack of specificity in transition dates and crop growth stages underscores the necessity of establishing a mapping relationship between these two sets of data. Although Figures 3 and 4 exclusively illustrate the spline-modeled phenological curve of TROPOMI SIF and MODIS EVI, this study conducted similar comparisons between the observed crop growth stages and phenologically extracted transition dates using the same methods. After reviewing four subfigures in Figures 3 and 4 and other instances, and matching each growth stage of maize and winter wheat with the phenologically characterized transition date that falls within the closest range, potential reconciliations of phenologically characterized transition dates with crop growth stages of maize and winter wheat are, respectively, presented in Tables 3 and 4. The term 'potential best' accounts for the uncertainty in the estimated accuracy of each crop growth stage in relation to the corresponding transition dates. The results show that different growth stages can map to one or none of the transition dates extracted by each phenological transition characterization method. Some transition dates are associated with several crop growth stages. Notably, when characterized with the GU method, the transition dates for different data exhibit varying mapping relationships with crop growth stages. In other words, the GU method may offer customized phenological characterizations for different datasets. The accuracy of these mapping estimations will be assessed in the following section.

**Table 3.** Potential best conciliations of phenological characterized transition dates with crop growing stages of maize.

|  | Threshold | Derivative | Curvature | Gu (ST) | Gu (In Situ) |
|---|---|---|---|---|---|
| Sowing | - | - | - | - | Upturn |
| Emerged | - | - | Greenup | Upturn | - |
| 5th leaf | - | - | Greenup | Upturn | Stabilization |
| Jointing | SOS | SOS | Greenup | Stabilization | Stabilization |
| Tasseling and silking | POS | POS | Maturity | Stabilization | Downturn |
| Milk | EOS | EOS | Senescence | Downturn | Recession |
| Maturity and harvest | EOS | EOS | Dormancy | Recession | - |

Note: "-" refers to no suited phenological characterized transition dates; 'ST' refers to satellite data.

**Table 4.** Potential best conciliations of phenological characterized transition dates with crop growing stages of winter wheat.

| | TR | DB | CU | GU (TROPOMI SIF) | GU (In Situ SIF) | GU (EVI) |
|---|---|---|---|---|---|---|
| Regreen | - | - | GU | UD | - | UD |
| Jointing | SOS | SOS | GU | - | UD | SD |
| Heading | POS | POS | MT | SD | SD | - |
| Milk | EOS | EOS | SN | DD | DD | RD |
| Ripening | EOS | EOS | DM | - | RD | RD |
| Harvest | - | - | DM | RD | RD | - |

Note: "-" refers to no suited phenological characterized transition dates.

### 3.2. Comparison between Time-Series Phenological Estimation Portfolios for Crop Growth Stages

The phenological estimation portfolio comprises various data sources, compositing methods and phenological transition characterization methods. To determine the optimal estimation portfolio within the phenological estimation framework, this section assesses the accuracy of each component. The raw results of accuracy assessments for all portfolios are included in the Appendix A. For this section, data have been selected based on the best accuracy within each portfolio category (e.g., $R^2$ of estimations using in situ SIF for maize growth stages in Figure 5a are derived from the best-performing in situ SIF data with various compositing methods and measurement times in the Appendix A).

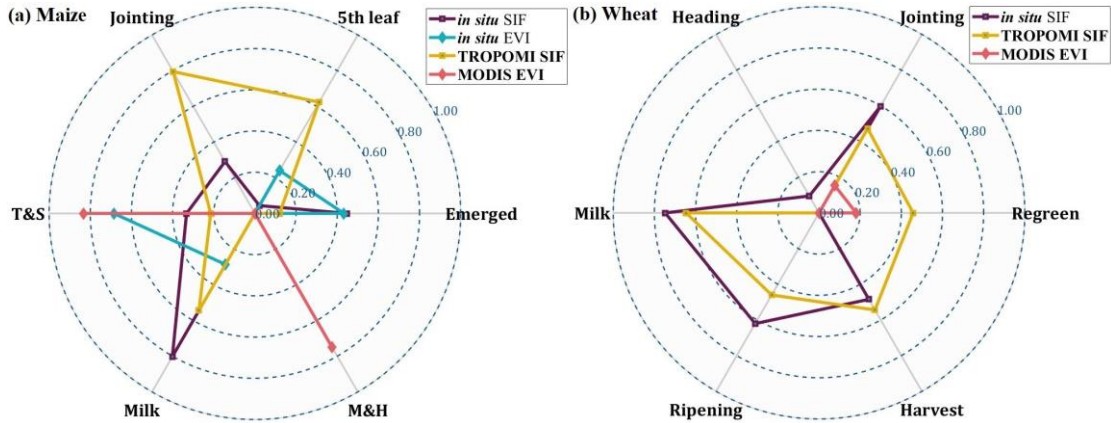

**Figure 5.** Accuracy assessment of estimations with different datasets. Panel (**a**) focuses on maize, while panel (**b**) pertains to winter wheat. The circular arcs in both radar charts represent $R^2$ for each estimation, ranging from 0 to 1. All the growth stages have been estimated with the listed datasets, with dots in the origin of both radar charts referring to $R^2$ of 0.

#### 3.2.1. SIF vs. EVI

As one of the main objectives of this study is to evaluate the suitability of SIF data for estimating crop growth stages within the agronomic framework, the initial step in this analysis compares the accuracy of estimations using SIF and EVI data. Figure 5 presents the accuracy of the estimation results for each crop growth stages with in situ SIF, TROPOMI SIF, in situ EVI and MODIS EVI.

For summer maize, as depicted in Figure 5a, EVI data exhibited superior accuracy in estimating tasseling and silking (T&S) as well as maturity and harvest (M&H) growth stages, especially for M&H, where SIF data cannot predict the date accurately. For these two growth stages, MODIS EVI exhibits lower $R^2$ than in situ EVI. Emerged growth stage is a special case where estimations with ground-based observations (in situ data) performed better than those with satellite data. Conversely, SIF data excelled in the estimation of the fifth leaf, jointing and milk, three of the four key growth stages, surpassing

the EVI data. Notably, TROPOMI SIF demonstrates favorable performance for fifth leaf and jointing stages, achieving accuracies approximately 0.6 and 0.8, respectively, and provided acceptable estimations for the milk stage with an accuracy exceeding 0.5. In summary, SIF data outperformed EVI data in identifying the key growth stages (three out of four) of maize, with the exception of T&S. TROPOMI SIF also proved useful for estimating most key growth stages of maize with acceptable accuracy.

Turning to winter wheat, Figure 5b indicates that SIF data could be effectively employed to estimate most growth stages of winter wheat, except for heading (similar to tasseling in maize, see above), while EVI data performed poorly. The absence of an in situ EVI dataset for winter wheat was attributed to its inadequate performance in growth stage identification. In the winter wheat growing season post-regreen, the crop's greenness remains stable. This leads to an unvarying EVI curve before the peak, posing challenges for phenological modeling and growth stage identification during this phase. Regarding the SIF data, in situ SIF performed slightly better than TROPOMI SIF for all of the KGSs (from jointing to maturity). But TROPOMI SIF, at the same time, can offer decent accuracy for the estimation of jointing, milk and maturity of winter wheat, which could also be adopted for the identification of regreen and harvest.

In summary, according to Figure 5, SIF datasets demonstrated superior performance compared to EVI datasets in identifying crop growth stages for both maize and winter wheat, especially for the KGSs and winter wheat. EVI data had limited utility in estimating winter wheat growth stages, while SIF data proved inadequate for T&S (in maize) and heading (in winter wheat) estimation. Additionally, TROPOMI SIF was found to offer accuracy with $R^2$ at approximately or over 0.6 for estimating most crop growth stages of maize and winter wheat.

### 3.2.2. Effect of Data-Measured Time Period on Estimation Accuracy

The impact of data-measured time periods ('Morning', 'Afternoon' or 'Whole-Day') on estimation accuracy was assessed in this section. The definition of these time periods is detailed in Section 2.3. Figure 6 illustrates the accuracy of estimation results for various crop growth stages using ground-based datasets.

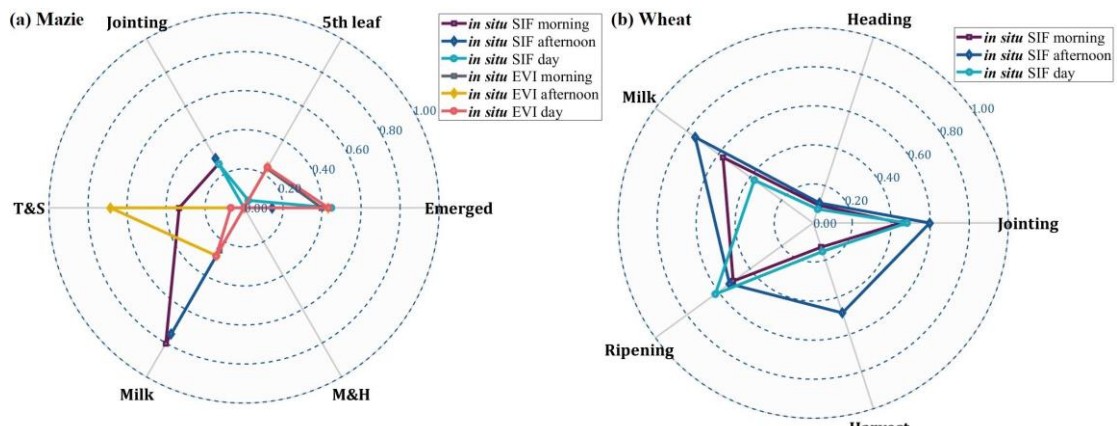

**Figure 6.** Accuracy assessment for crop growth stage estimations using different datasets. Subfigure (**a**) refers to maize, while subfigure (**b**) is focused on winter wheat. The circular arcs in both radar charts represent $R^2$ for each estimation, ranging from 0 to 1. The labels 'morning', 'afternoon' and 'day' in this figure correspond to the data-measured time periods of 'Morning', 'Afternoon' and 'Whole-Day', respectively.

For summer maize, the results shown in Figure 6a indicate a complex pattern when considering different data-measured time periods. In terms of in situ SIF data, 'Morning' measurements generally perform better than data from other time periods. However, there are exceptions, notably for T&S, where only the 'Morning' dataset provides an accuracy

of approximately 0.37, which is still considered unacceptable. The estimation accuracy of SIF measured during 'Afternoon' is slightly higher than that of SIF during 'Morning' for jointing, while the opposite is true for T&S and milk. Regarding in situ EVI data, the results in Figure 6a suggest that 'Afternoon' measurements are generally the most suitable for maize growth stage estimation. EVI data from 'Afternoon' significantly outperform data from other time periods for estimating T&S, while offering similar accuracy for the estimation of other growth stages.

As for winter wheat, Figure 6b demonstrates that in situ SIF data measured during 'Afternoon' consistently provide outstanding performance in the estimation of almost all winter wheat growth stages, except for maturity. For Maturity, SIF data from 'Whole-Day' offer slightly better accuracy, while SIF data from the 'Afternoon' still yield a decent accuracy of approximately 0.53.

In summary, for maize, the choice of data-measured time period has a mixed impact on estimation accuracy, with 'Morning' being generally better for in situ SIF data, and 'Afternoon' being more suitable for in situ EVI data. For winter wheat, 'Afternoon' measurements of in situ SIF data consistently offer the most accurate estimation results for most growth stages. To be noticed, Figure 6b does not include regreen growth stage because in situ SIF could not be utilized for estimation if it (Figure 5b). The absence of in situ EVI data was due to its extremely low accuracy elaborated in the previous section.

### 3.2.3. Compositing Methods

The concept of compositing methods pertains to the manner in which data are combined—whether by taking the maximum or average value—and the duration of time intervals between each pair of compositing dates during the observation period. This section aims to assess how different compositing methods influence estimation results and identify, where possible, the most effective compositing method. Notably, the MODIS EVI product used in this study was already composited into 16-day intervals using the MVC method. Consequently, satellite EVI data are not considered in this section. Furthermore, in situ EVI data were excluded for winter wheat due to its documented performance issues, as previously discussed.

This section begins by investigating the impact of different time interval lengths between compositing dates on estimation accuracy. In Figure 7, it presents the distribution of estimations for maize and winter wheat KGSs using SIF datasets with varying composited intervals, categorized accordingly. Each observation (OB) point, represented by a red dot, on the chart illustrates the distribution of observed KGSs for each target crop, and this pattern applies to the other points as well. The distance between each point and the origin indicates the precision (standard deviation) of estimations using the corresponding time interval, while the distance between each point and the observation point signifies the accuracy (the root-mean-square-error, RMSE), represented by the point's color. Additionally, the 'Correlation' value on the arc also reflects the accuracy (R) of each estimation. As discussed in Section 3.2.1, EVI datasets show limited capability in estimating all KGSs accurately when treated as a complete dataset, although they may perform well for specific KGSs (e.g., T&S of maize, as seen in Figure 6). Consequently, EVI datasets have been excluded from this analysis. In general, the standard deviations (STDs) of most ground-based observations are approximately 25 (except for the area observation of winter wheat), which corresponds to the average lengths of the key growth periods for both crops. Likewise, the STDs of most estimations fall within the same scale, except for maize with TROPOMI SIF, where the STD is approximately 20. The RMSEs for most estimations are lower than 20, and RMSEs for estimations of winter wheat and maize with in situ SIF are even below 10. This suggests that most SIF datasets are capable of providing estimations with good accuracy and precision, with estimations for winter wheat generally exhibiting slightly higher accuracy than those for maize. Comparing estimations with TROPOMI SIF, most of which provide correlations (R) higher than 0.7, indicating a decent level of accuracy. Estimations with in situ SIF consistently offer correlations approximate 0.9 or even higher,

highlighting their remarkable accuracy. Regarding the ideal compositing intervals, it is important to choose not only the ones with lowest RMSEs, but also considering the STDs and correlations. It is advisable to choose 15 days and 7 days when estimating KGSs for maize and winter wheat, respectively, using in situ SIF. For TROPOMI SIF, the optimal compositing intervals are 7 days for maize KGSs, while no compositing (1d) is recommended for winter wheat KGSs. Notably, the divergence in ideal compositing intervals for the two crops across varying compositing methods underscores the challenge of finding a single compositing method suitable for accurately estimating KGSs for each dataset.

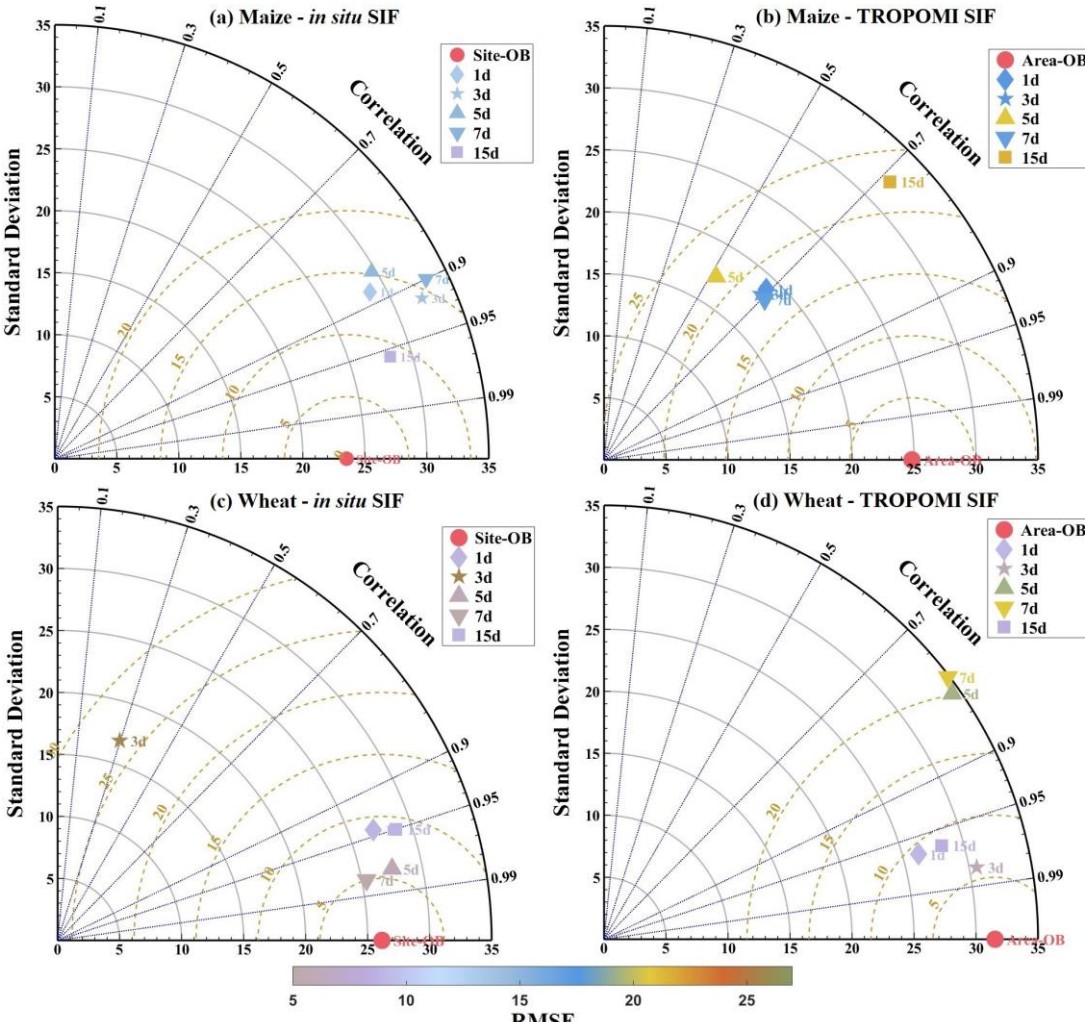

**Figure 7.** Taylor diagrams for estimations of key growth stages (KGSs) estimations using varying lengths of composited intervals. From subplot (**a**–**d**), it illustrates the distribution of estimations of maize KGSs using in situ data (SIF and EVI) and TROPOMI SIF and winter wheat KGSs using in situ SIF and TROPOMI SIF, respectively. 'OBs' in the legend represent the ground-based observations, the reserved dataset, of each category. In this figure, each dot corresponds to a specific data source with the corresponding length of composited intervals, while the distance between each dot and the origin represents the standard deviation of each data source. The angle of each dot in this angular coordinate system refers to the 'Correlation' (R), while the distance between each dot and the reserved point (observation) is the RMSE of the estimation with each data source, which is also visualized by the color of each dot according to the color bar.

The choice of composited value is another crucial aspect of the compositing method. In the case of maize growth stage estimation using in situ SIF, as depicted in Figure 8a, it is evident that MVC outperforms AVC, as indicated by consistently higher $R^2$ for each

crop stage. In the case of in situ EVI, MVC also performs better than AVC, although the advantage is somewhat less pronounced, with AVC being notably better for the emerged stage. However, the assessment result for TROPOMI SIF is less clear-cut. AVC performs better for the emerged, milk and, especially, jointing stages, while MVC is more suitable for the V5 and T&S stages. Turning to winter wheat, as shown in Figure 8b, MVC consistently outperforms AVC for all stages. However, when considering TROPOMI SIF, the situation becomes somewhat nuanced. AVC yields slightly more accurate estimations than MVC for most stages, except maturity stage. In summary, it appears that MVC is generally more suitable for ground-based datasets, while AVC might be marginally more appropriate for TROPOMI SIF datasets in the context of crop growth stage estimation.

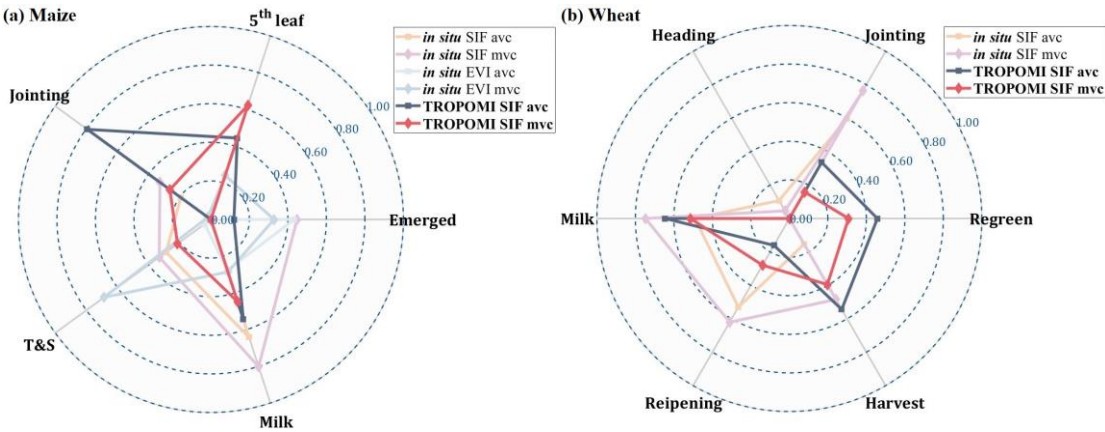

**Figure 8.** Accuracy assessment of estimations with different composited values (MVC and AVC). (**a**) focuses on maize, while (**b**) pertains to winter wheat. The circular arcs in two radar charts are $R^2$ of each estimation, ranging from 0 to 1. In this figure, the lines with square nodes indicate the use of AVC, while the lines with diamond nodes represent MVC.

### 3.2.4. Phenological Modeling and Transition Characterization

In this critical section, the impact of phenological modeling and transition characterization methods on the accuracy of our crop growth stage identification framework is thoroughly examined. Notably, these choices can significantly influence the precision of our entire approach. Additionally, EVI datasets were excluded from comparisons concerning winter wheat due to their previously established performance limitations.

Figure 9b,d clearly illustrate that, for both EVI and TROPOMI SIF datasets, the models perform best when ranked as follows: the Gu model, Spline model and Beck model. However, when applied to in situ SIF data (Figure 9a), the situation differs between maize and winter wheat. For maize, the model ranking aligns with the general trend, but for winter wheat, the Beck model yields the highest accuracy, followed by the Gu model and the Spline model. Moving to the realm of phenological transition characterization methods, in situ EVI and TROPOMI SIF tend to favor the CU (curvature-based) method, whereas the MODIS EVI dataset leans towards the TR (threshold-based) or DB (derivative-based) methods. Notably, regarding in situ SIF in Figure 9a, none of the four phenological transition characterization methods exhibit significant superiority. As for the absolute average value of $R^2$, in situ SIF dataset exhibits superiority over the other three datasets. This superiority can be attributed to the more adequate measurements compared with TROPOMI SIF and advantage of SIF data over EVI data in terms of estimation accuracy as discussed in previous sections.

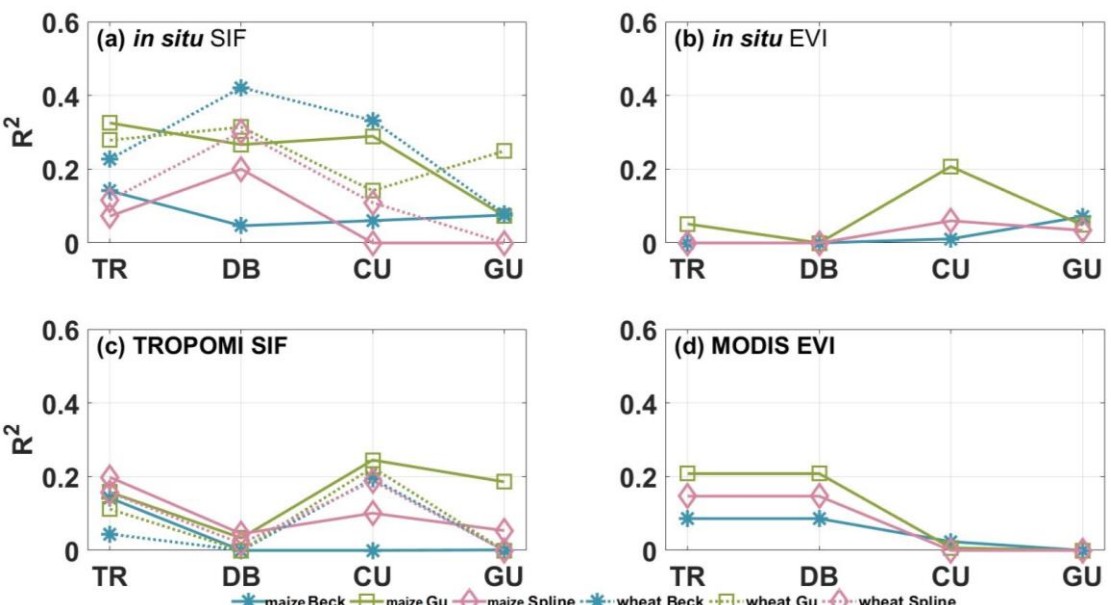

**Figure 9.** Average of $R^2$ for key growth stage identification accuracy using different phenological modeling and transition characterization datasets. Each subfigure uses distinct colors to represent specific phenological modeling methods, with solid lines for maize and dotted lines for winter wheat results. The *x*-axis lists the four phenological transition characterization methods.

The average $R^2$ values presented in Figure 8 are relatively lower compared to other figures. This is because the average $R^2$ here represents the mean of $R^2$ values for each individual key growth stage (KGS) using different phenological modeling and transition characterization datasets. As illustrated in Figures 5, 6 and 8, estimations with a specific dataset may exhibit decent accuracy for certain crop growth stages while showing lower accuracy for others. Consequently, this variability contributes to a relatively lower average $R^2$. Furthermore, the relatively low average $R^2$ in Figure 9 underscores the complexity of selecting an optimal approach suitable for all situations. The challenge lies in the fact that different phenological modeling and transition characterization methods may perform differently across various crop growth stages, making it difficult to identify a one-size-fits-all solution.

In summary, the Gu model appears to be the most suitable choice for most crop key growth stage (KGS) estimations, with the notable exception of the in situ SIF curve of winter wheat, where the Beck model could yield superior results. However, there is not a one-size-fits-all solution when it comes to phenological transition characterization methods. The choice of optimal approach is complex, which often involves a tailored combination of models, methods and even source data for each specific KGS. This complexity may, in part, explain the relatively low mean $R^2$ values observed in both this section and the previous sections, as the average $R^2$ alone may not fully capture the nuanced performance of our phenological metrics in estimating KGSs.

### 3.3. Evaluation of Best Time-Series Phenological Estimation

This section delves into the comprehensive evaluation of our time-series phenological estimation portfolio, encompassing preprocessing steps, modeling techniques, and transition characterization methods. The aim is to gauge the accuracy of the best phenological estimation portfolio. Given the objective of evaluating the efficacy of SIF data in crop growth stage estimation within the agronomic framework, and the observations made in Section 3.2.1, highlighting the advantages of SIF data over EVI data, this section concentrates solely on SIF data in this evaluation.

The best portfolio comprises a harmonious amalgamation of several components, including the data-measured time period (if necessary), compositing methods, phenologi-

cal curve-fitting models and transition characterization methods, all chosen to optimize accuracy for each (KGS). However, selecting a universal compositing or transition characterization method suitable for all KGSs is fraught with difficulty. Therefore, this study only chose the appropriate data-measured time period (e.g., 'Morning' for maize and 'Afternoon' for winter wheat, according to the results elaborated in Section 3.2.2 and Figure 5) and the Gu model as the better-picked scheme (named after 'bp'), together adopted with different compositing methods and transition characterization methods. These methods adopted by the 'bp' scheme have been listed in Table 5. Notably, the measured times and compositing methods of in situ SIF for maize and winter wheat are consistent with the results elaborated in Sections 3.2.2 and 3.2.3.

**Table 5.** Methods, apart from Gu model for curve-modeling, adopted by the 'bp' scheme for KGS estimation in this study.

| Crop | Data | Growth Stage | Measured-Time | Compositing Method | Characterization Method |
|---|---|---|---|---|---|
| Maize | In situ SIF | V5 | Morning | 5d-MVC | GU |
| | | JT | Morning | 5d-MVC | CU |
| | | T&S | Morning | 5d-MVC | TB |
| | | MK | Morning | 5d-MVC | CU |
| | TROPOMI SIF | V5 | - | 7d-MVC | GU |
| | | JT | - | 15d-AVC | TB |
| | | T&S | - | 7d-MVC | GU |
| | | MK | - | 15d-AVC | CU |
| Winter wheat | In situ SIF | JT | Afternoon | 7d-MVC | TB |
| | | HD | Afternoon | 7d-MVC | TB |
| | | MK | Afternoon | 7d-MVC | CU |
| | TROPOMI SIF | JT | - | 1d | DB |
| | | HD | - | 1d | TB |
| | | MK | - | 1d | CU |

Figures 10 and 11 offer insights into the accuracy of our estimations for maize and winter wheat, respectively, utilizing SIF data processed by both the best possible portfolios and the 'bp' schemes. Notably, there are only three data points for each growth stage in Figure 11a,b, as explained in Section 2, due to the limited observations available. The distribution of dots along the horizontal axis reflects the precision of the ground-observed data, while their distribution along the vertical axis indicates the precision of the estimation results. Moreover, the distance of these dots from the fitting line serves as an indicator of the accuracy of the estimation. Notably, the $R^2$ presented in these figures quantify the accuracy of the estimation relative to the fitting line, rather than the ground observations. However, the observations can be integrated with the estimations using the provided formulas in the figures. Thus, the $R^2$ still serve as a comprehensive measure of the overall accuracy of our KGS estimation framework.

In general, all estimation results exhibit remarkable distinctions in both precision and accuracy, underscoring the robust capabilities of SIF data in KGS estimation. Additionally, the accuracy of estimations involving TROPOMI SIF data slightly surpasses that of estimations utilizing in situ SIF data. With regard to the 'bp' schemes, the accuracy gap between estimations processed by best portfolios and 'bp' schemes is very narrow. Even the lowest 'bp' accuracy, standing at 0.81 in $R^2$ (in situ SIF—'bp' for maize in Figure 10b), underscores the substantial similarity between the 'bp' schemes and the best portfolios. Consequently, our estimation framework can be streamlined with the 'bp' schemes, preserving accuracy and reducing complexity.

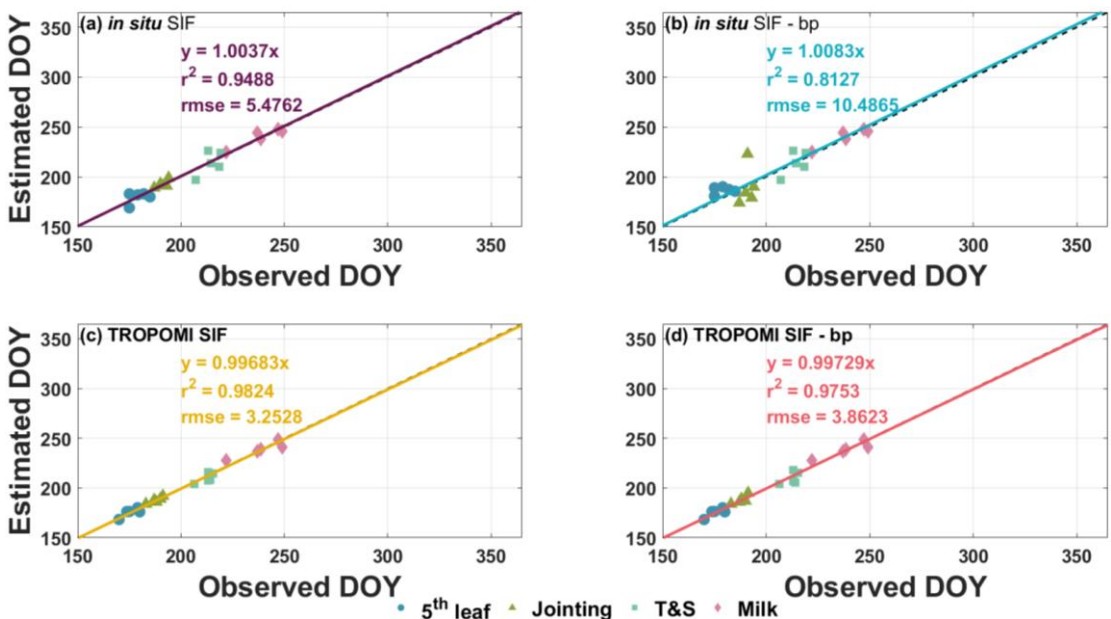

**Figure 10.** Accuracy assessment of maize growth stage estimations with in situ SIF, TROPOMI SIF and the better-picked ('bp') schemes for each dataset. Each dot represents one growth stage from one year. The dot line and solid line correspond to the 1:1 line and fitting line of the estimation, respectively. Subfigures (**a**,**c**) depict the estimations processed by the best possible estimation portfolios, while subfigures (**b**,**d**) illustrate the estimations processed by the 'bp' schemes.

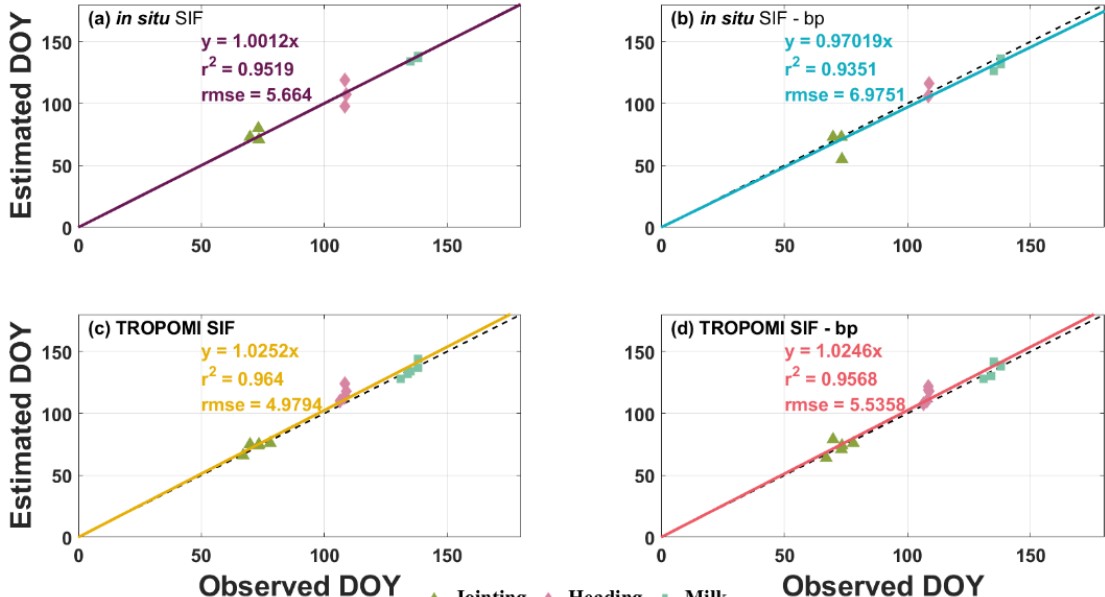

**Figure 11.** Accuracy assessment of winter wheat growth stage estimations using in situ SIF, TROPOMI SIF and the better-picked ('bp') schemes for each dataset. Each dot within the figure represents one growth stage from a given year. The dot line and solid line correspond to the 1:1 line and fitting line of the estimation, respectively. (**a**,**c**) present estimations processed by the best possible estimation portfolios, while (**b**,**d**) illustrate the estimations processed by the 'bp' schemes. Please note that the *x*-axis range has been adjusted to accommodate the winter wheat growing season.

## 4. Discussion

### 4.1. Capability of SIF Data in the Crop Growth Stage Estimation Framework

The objective of this study was to assess the capability of SIF data within a crop growth stage estimation framework, particularly under the context of agronomic practices. A site on the North China Plain served as a case study to examine ground and satellite-based SIF data, comparing their performance with traditional vegetation indices.

While prior research has explored the relationship between SIF and gross primary production, it predominantly focused on land surface phenology, typically limited to the start or end of the growing season [18,45,46]. In contrast, cropland systems involve complexities, demanding more precise and detailed information for crop growth monitoring. Some studies emphasized crop growth monitoring and the importance of aligning remote sensing transition dates with specific crop growth stages within an agronomic framework. [6,7,42,66,68,77]. Notably, SIF data has not been rigorously evaluated within this framework. Additionally, previous works often treated distinct crop growth stages as equal events without considering their varying importance based on crop physiology and economic factors [6,66]. However, the importance of different crop growth stages is not the same based on the physiological characters of crops and the economic factors of agricultural activities. Therefore, this study evaluates the potential of SIF data in estimating KGSs within this comprehensive framework.

The findings reveal that, in the context of maize, SIF data outperforms EVI data for the identification of most KGSs, except for T&S (Figure 5). Conversely, EVI data excels in detecting T&S, suggesting the possibility of combining SIF and EVI data for more accurate maize growth stage estimation. In contrast, winter wheat exhibits different growth dynamics, with the ability to sustain greenness from regreen to maturity, minimizing EVI data fluctuations. SIF data, which reflects photosynthetic activity, remains unaffected and performs well in winter wheat growth stage estimation as Figure 5 illustrates [27,78,79]. Moreover, the comparison between ground-based and satellite data demonstrates that TROPOMI SIF provides superior or, at the very least, reasonably accurate estimations.

In summary, considering both the alignment between remote sensing transition dates and specific crop growth stages and the importance of different crop growth stages, this study evaluated SIF data integrated into a remote sensing crop growth stage estimation framework. Compared with EVI data, which reflects the greenness of monitoring crops, SIF data have superior performance on winter wheat growth stages estimation with the ability to reflect photosynthetic activity. SIF data also offers more accurate estimations for maize. Thus, SIF data are capable for crop growth stage estimation. Notably, TROPOMI SIF data exhibits the potential to perform crop growth stage identification at a regional scale without compromising accuracy, highlighting its utility in agricultural monitoring applications.

### 4.2. Evaluation of Elements in The Estimation Portfolio

In the assessment of elements within the crop growth stage estimation portfolio, this study delves into the combination of factors, including data-measured time periods, compositing methods, phenological curve-fitting models and transition characterization methods. While previous research has acknowledged these elements as potential solutions for specific situations, there remains a dearth of comprehensive comparisons among them [66,68].

Commencing with the selection of data-measured time periods, the findings reveal divergent results for maize and winter wheat growth stage estimation using SIF data (Figure 6). 'Afternoon' emerges as the optimal measurement time for winter wheat, while 'Morning' proves superior for maize. This discrepancy can be attributed to the midday depression in photosynthesis that the intensity of photosynthesis have been found reduced during the heat wave at mid of the day [80]. At the same time, this depression in photosynthesis was tracked by SIF observation for its sensitivity with the gross primary production loss [21,46]. This sensitivity of SIF underscores the influence of the midday depression

on constructing phenological time-series curves. The difference between winter wheat and maize is linked to variations in midday temperatures during their respective growing seasons in Shangqiu. Additionally, 'Afternoon' emerges as the preferred measurement time for EVI, indicating EVI's insensitivity to midday photosynthetic depression.

The impact of compositing methods on phenological monitoring has been discussed in prior studies, with appropriate methods shown to reduce systematic errors in estimations by mitigating sensitivities to factors such as canopy structure, chlorophyll content and biomass [71,81,82]. Compositing methods encompass composited values and lengths of composited intervals. In terms of the length of composited intervals, this study suggests that for datasets utilized in maize KGS estimation, an interval of approximately 7 days is most suitable (Figure 7). Conversely, for datasets employed in winter wheat KGS estimation, the ideal composited interval ranges from 1 day to 3 days. However, regarding composited values, it remains challenging to draw a definitive conclusion, with a vague trend indicating that MVC and AVC are preferable for EVI and SIF datasets, respectively (Figure 8). The findings concerning the impact of compositing methods on crop growth stage estimation lack the robustness required for general applicability.

Furthermore, this study explores the combinations of phenological curve-fitting models and transition characterization methods as potential sources of diversity in phenological event detection [6,42,68]. Yet, it is essential to streamline the phenological estimation framework, avoiding redundancy. As shown in Figure 9, the study attempts to identify an acceptable combination of models and methods to simplify the process, with Gu-based models demonstrating superior performance in most cases, if supplied with enough data [73]. However, the advantages of different phenological transition characterization methods remain unclear, as accuracy assessment results are distributed randomly. As a result, a robust combination suitable for general scenarios could not be ascertained in this case study. The combination of land surface model and agronomy crop model has been proved capable for improving maize growth processes simulation [83]. Remote sensing observations covering heat, water and photosynthesis assimilated into crop models could also improve crop phenological transition characterization. Thus, future research conducted on a larger scale with more extensive datasets assimilated into crop models may shed light on this issue.

In summary, this study scrutinizes individual components of the crop growth stage estimation framework, aiming to identify a unified portfolio for simplification. While some aspects of the portfolio, such as data-measured time periods, composited values and curve-fitting models, are discernible, others remain open to recommendation. Encouragingly, the results indicate significant precision and accuracy in the estimation of the best time-series phenological estimation portfolio and 'bp' schemes (Figures 10 and 11). Future studies, utilizing higher-resolution data and expanding the scope to encompass larger research regions, hold the potential to enhance the framework further.

## 5. Conclusions

This study delved into the capacity of SIF data, obtained from both ground-based and satellite measurements, to delineate crop growth stages, juxtaposed against EVI data, within the confines of our study site. It pioneered the harmonization of time-series phenological attributes with crop growth stages, thus underscoring the imperative need for aligning these two domains to effectively reconstruct their mapping relationship. The investigation unequivocally affirmed the effectiveness of SIF data within the phenological estimation framework for discerning the growth stages of both maize and winter wheat. SIF data emerged as the superior choice over EVI data, displaying marked advantages in terms of accuracy, robustness and sensitivity to phenological events. The remarkable precision achieved through the processing of SIF data, employing the best portfolio and 'bp' schemes, opens up promising avenues for further applications of SIF data in the domain of crop growth stage identification. Nevertheless, while this study offers valuable insights, it remains evident that an all-encompassing portfolio with fixed processing methods has yet to be fully unveiled. The pursuit of such a portfolio beckons further research

endeavors, propelling the field closer to a unified and optimized approach for crop growth stage identification.

**Author Contributions:** Conceptualization, Y.Z. and Y.H.; methodology, Y.H.; software, Y.H.; validation, Y.Z. and Y.H.; formal analysis, Y.H.; investigation, Y.H.; resources, L.W., L.P., D.D., G.W., Z.L. and Y.W.; data curation, L.W., L.P., D.D., G.W., Z.L. and Y.W.; writing—original draft preparation, Y.H.; writing—review and editing, Y.Z. and Y.H.; visualization, Y.H.; supervision, Y.Z., Z.L. and Z.Z.; project administration, Y.Z.; funding acquisition, Y.Z. All authors have read and agreed to the published version of the manuscript.

**Funding:** This research was supported by the National Natural Science Foundation of China (42125105, 42071388).

**Data Availability Statement:** The ungridded TROPOMI SIF740 product from 2018 to 2022 adopted in this study was sourced from ftp://fluo.gps.caltech.edu/data/tropomi/ungridded/ (accessed on 8 September 2023) (for additional details, refer to: https://doi.org/10.22002/D1.1347 (accessed on 8 September 2023)). The MODIS product utilized in this study is downloaded from the official website of United States Geological Survey at LP DAAC—MOD13C1 (usgs.gov) (accessed on 8 September 2023) and LP DAAC—MYD13C1 (usgs.gov) (accessed on 8 September 2023).

**Acknowledgments:** We would thank Qian Zhang, Ji Li and other researchers from Nanjing University for the installation of SIF and EC Flux equipment. Our appreciation also goes to Pengju Wu and other researchers from ShangQiu station of Farmland Irrigation Research Institute of Chinese Academy of Agricultural Sciences for their valuable contributions in the field measurements.

**Conflicts of Interest:** The authors declare no conflict of interest.

## Appendix A

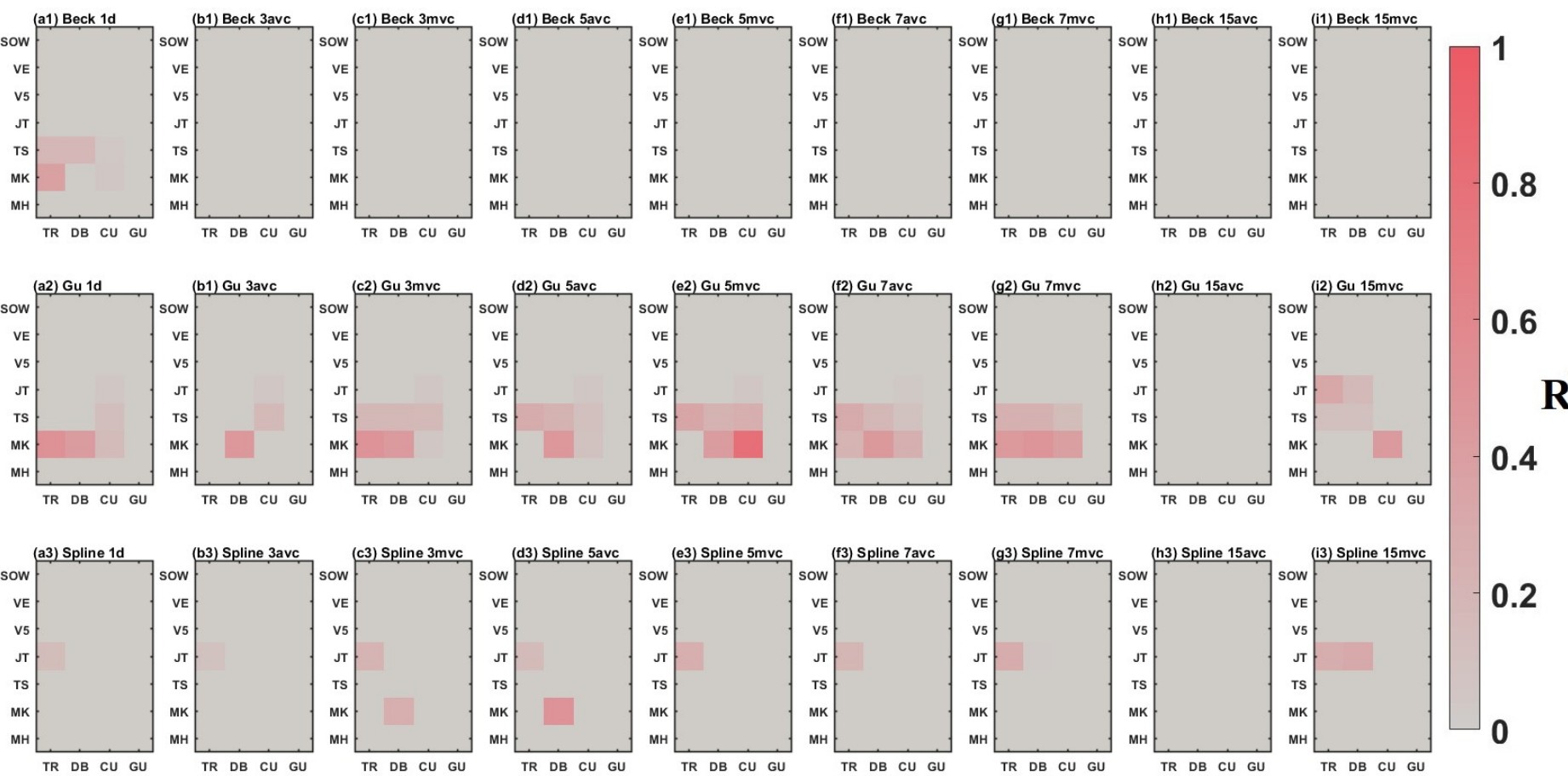

**Figure A1.** $R^2$ for each maize growth stage estimations, from 2018 to 2022, using in situ SIF measured from 'morning' with varying estimation portfolios. For each growth stage estimation, the greyer sector presents the lower $R^2$ for the estimation, while the redder sector presents the higher $R^2$.

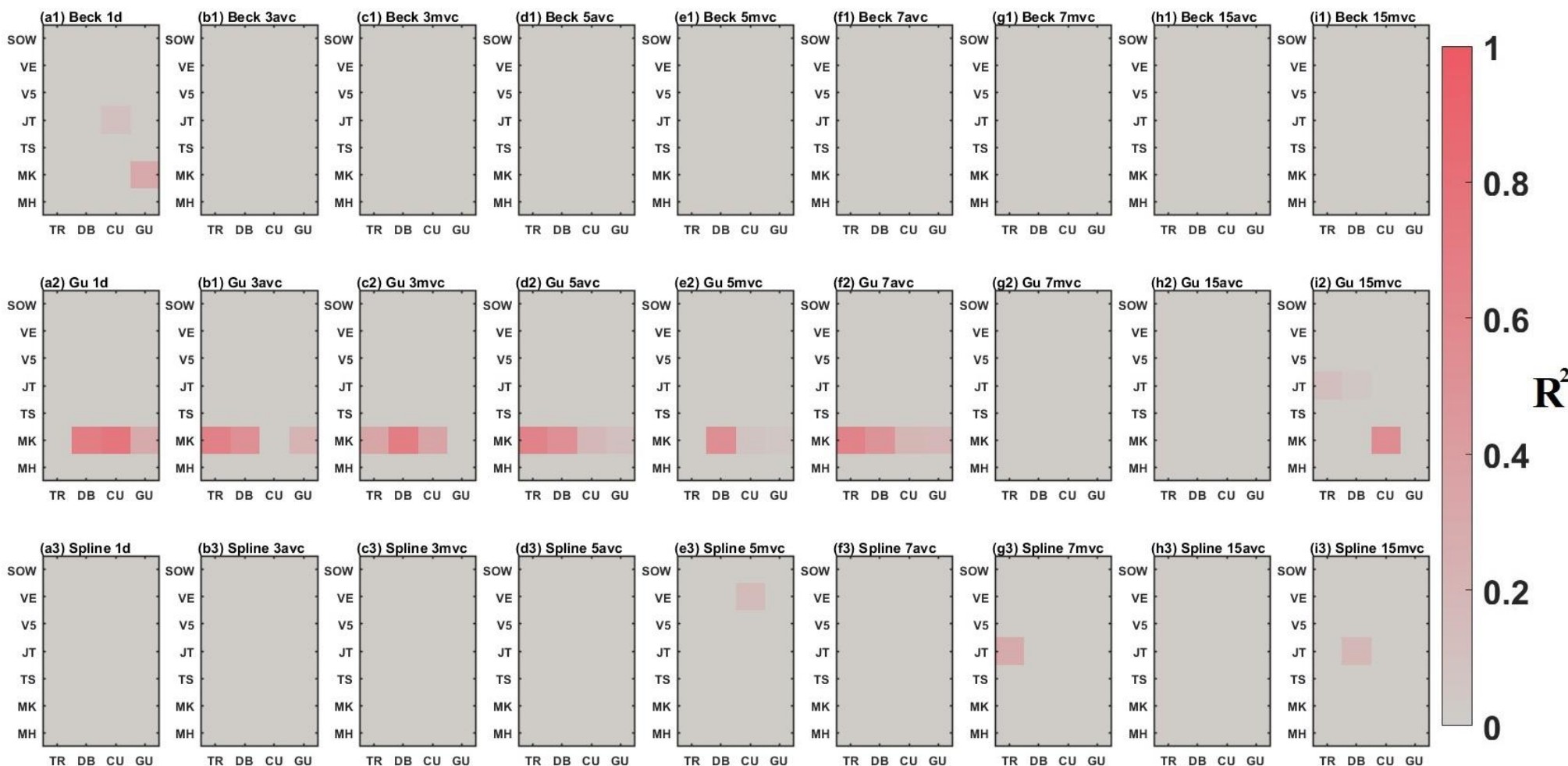

**Figure A2.** $R^2$ for each maize growth stage estimations, from 2018 to 2022, using in situ SIF measured from 'afternoon' with varying estimation portfolios. For each growth stage estimation, the greyer sector presents the lower $R^2$ for the estimation, while the redder sector presents the higher $R^2$.

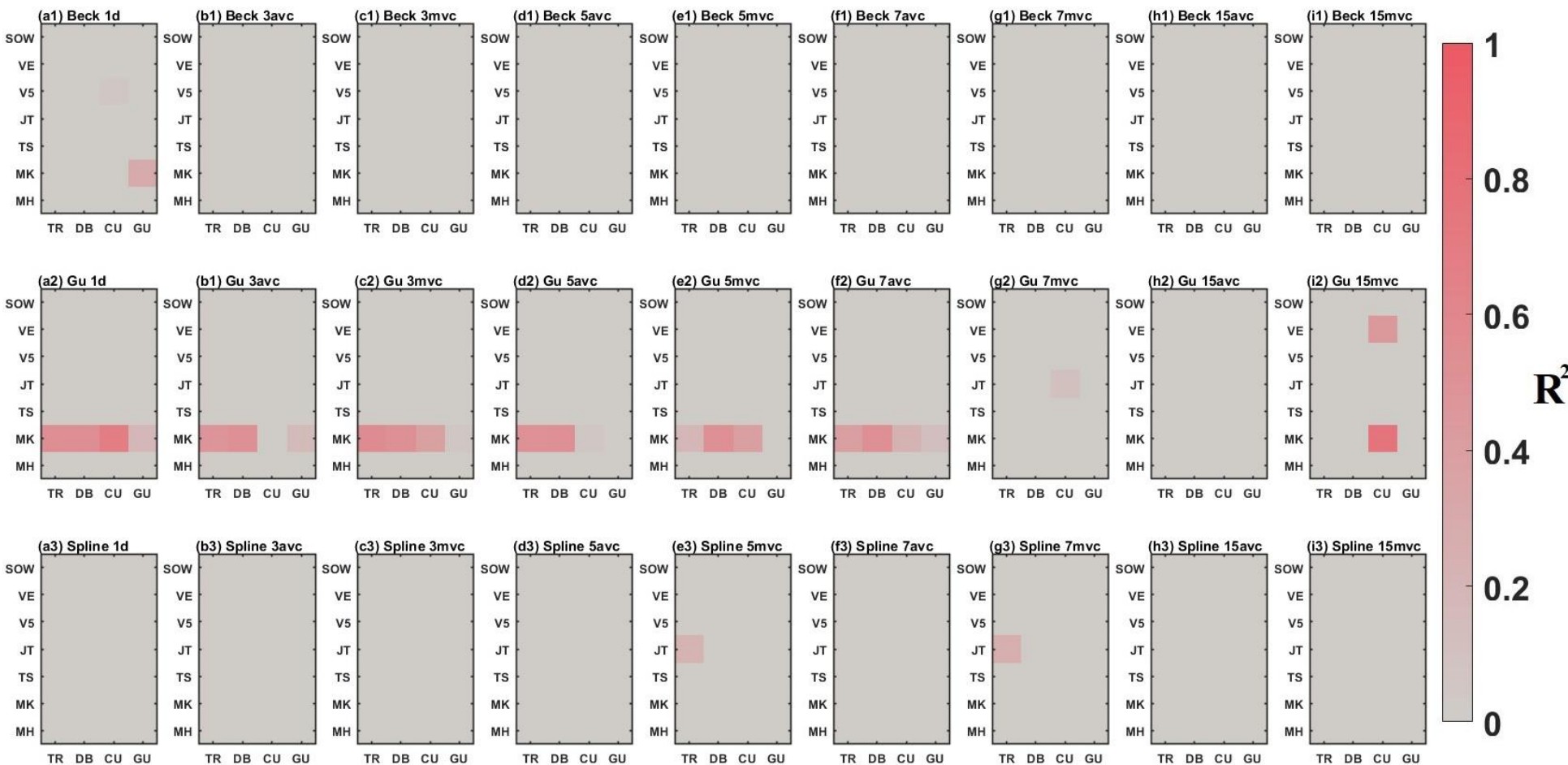

**Figure A3.** $R^2$ for each maize growth stage estimations, from 2018 to 2022, using in situ SIF measured from 'whole-day' with varying estimation portfolios. For each growth stage estimation, the greyer sector presents the lower $R^2$ for the estimation, while the redder sector presents the higher $R^2$.

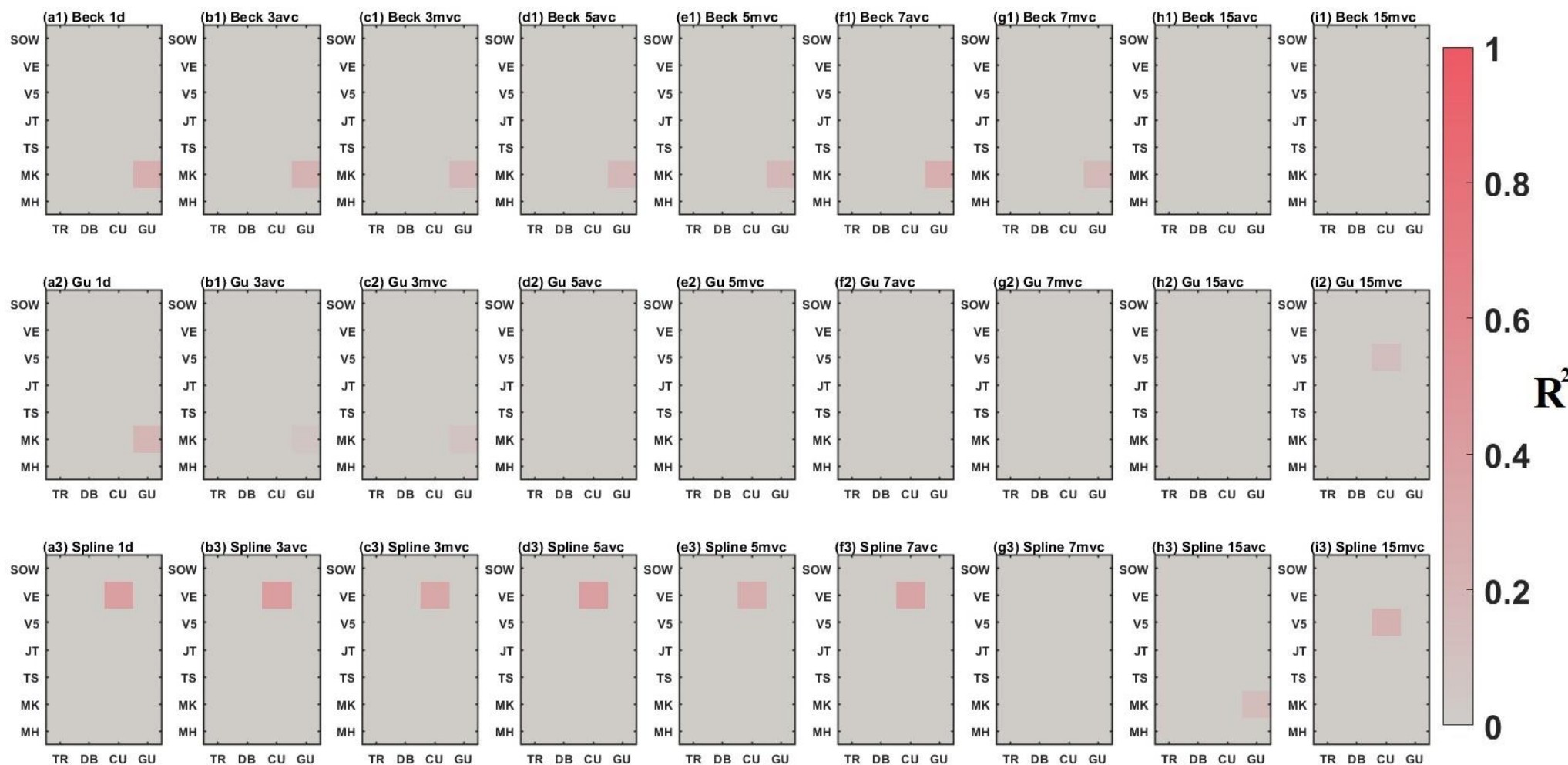

**Figure A4.** $R^2$ for each maize growth stage estimations, from 2019 to 2022, using in situ EVI measured from 'morning' with varying estimation portfolios. For each growth stage estimation, the greyer sector presents the lower $R^2$ for the estimation, while the redder sector presents the higher $R^2$.

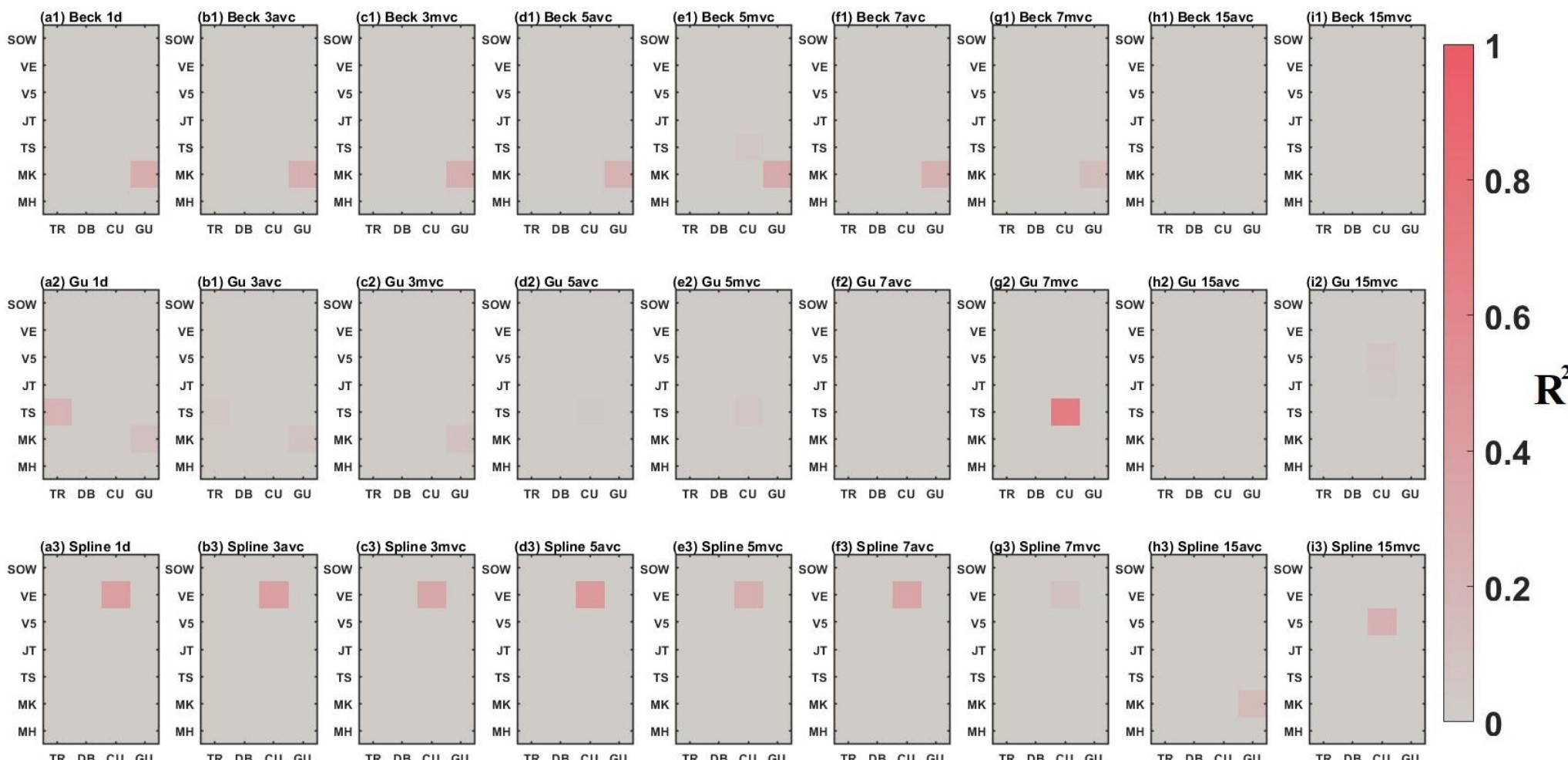

**Figure A5.** $R^2$ for each maize growth stage estimations, from 2019 to 2022, using in situ EVI measured from 'afternoon' with varying estimation portfolios. For each growth stage estimation, the greyer sector presents the lower $R^2$ for the estimation, while the redder sector presents the higher $R^2$.

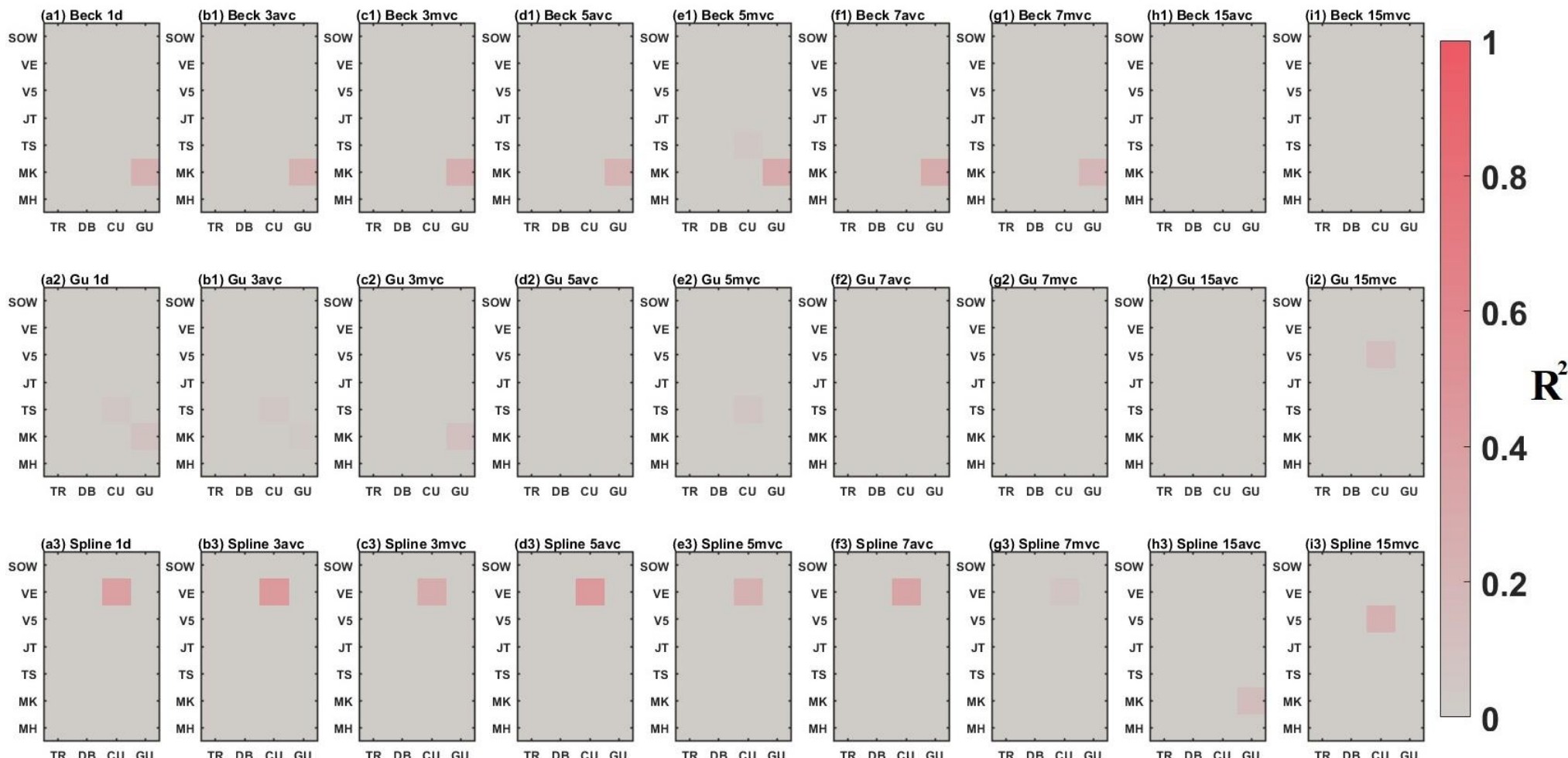

**Figure A6.** $R^2$ for each maize growth stage estimations, from 2019 to 2022, using in situ EVI measured from 'whole-day' with varying estimation portfolios. For each growth stage estimation, the greyer sector presents the lower $R^2$ for the estimation, while the redder sector presents the higher $R^2$.

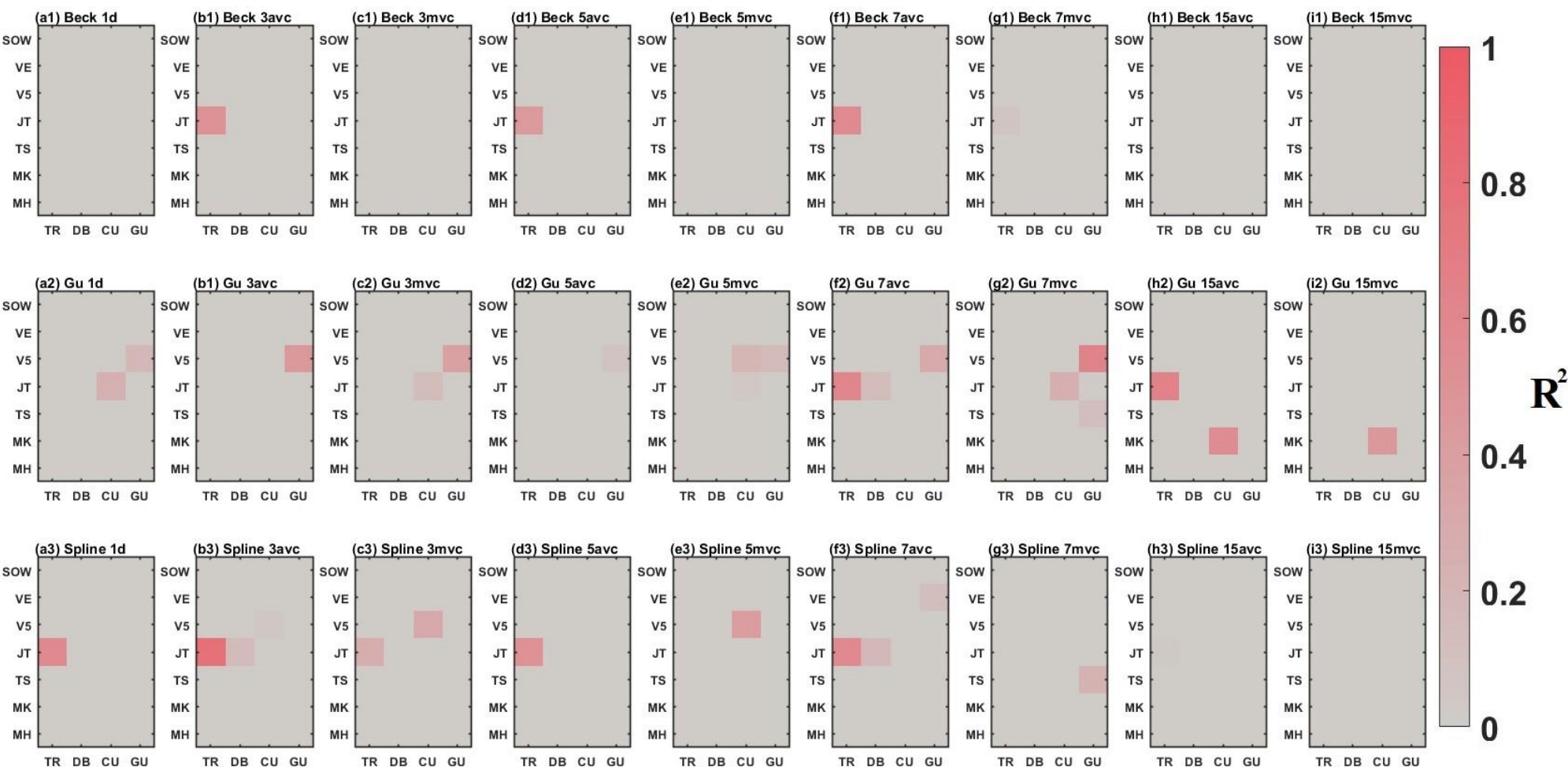

**Figure A7.** $R^2$ for each maize growth stage estimations, from 2018 to 2022, using TROPOMI SIF with varying estimation portfolios. For each growth stage estimation, the greyer sector presents the lower $R^2$ for the estimation, while the redder sector presents the higher $R^2$.

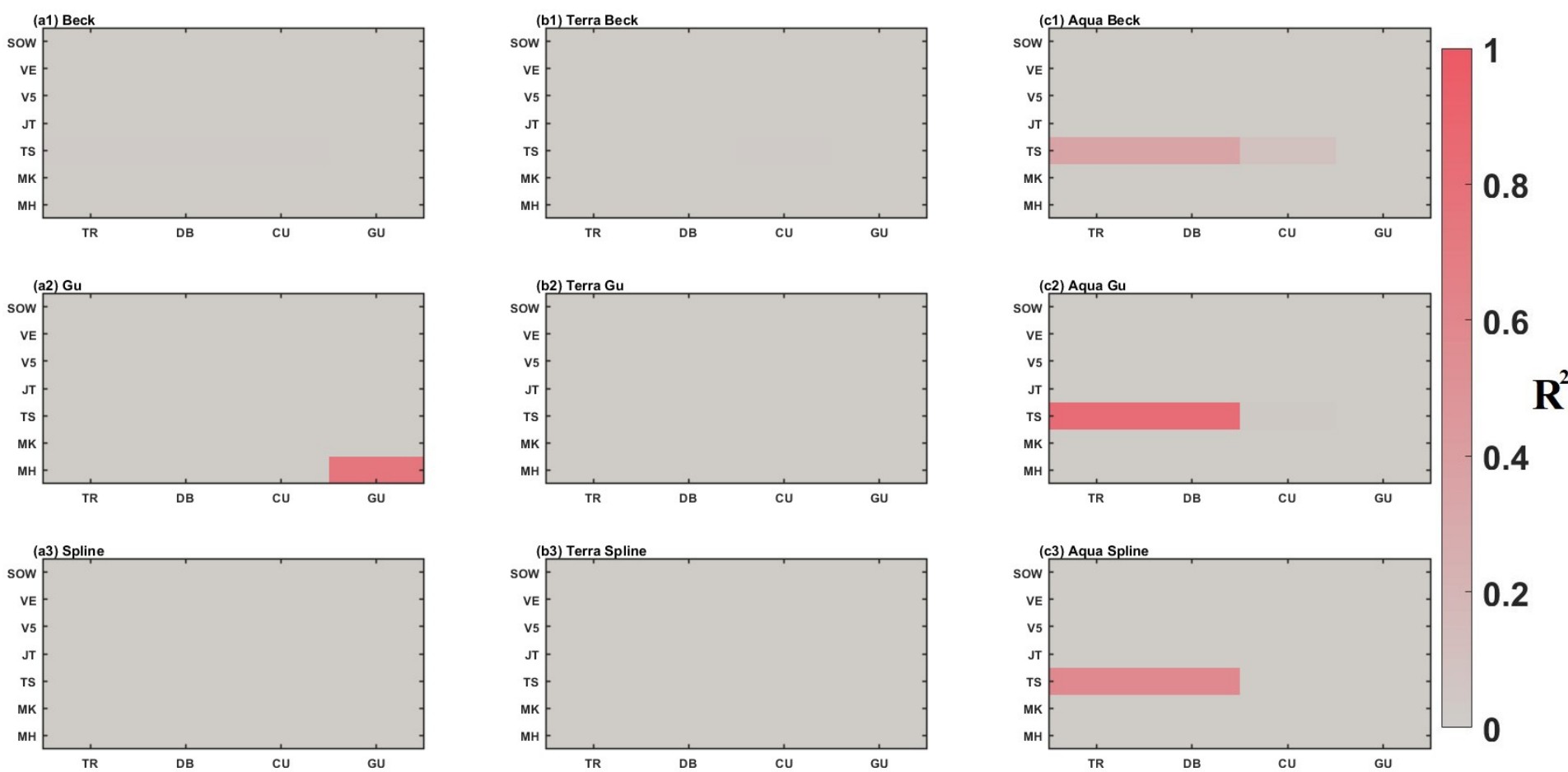

**Figure A8.** $R^2$ for each maize growth stage estimations, from 2018 to 2022, using MODIS EVI with varying estimation portfolios. For each growth stage estimation, the greyer sector presents the lower $R^2$ for the estimation, while the redder sector presents the higher $R^2$.

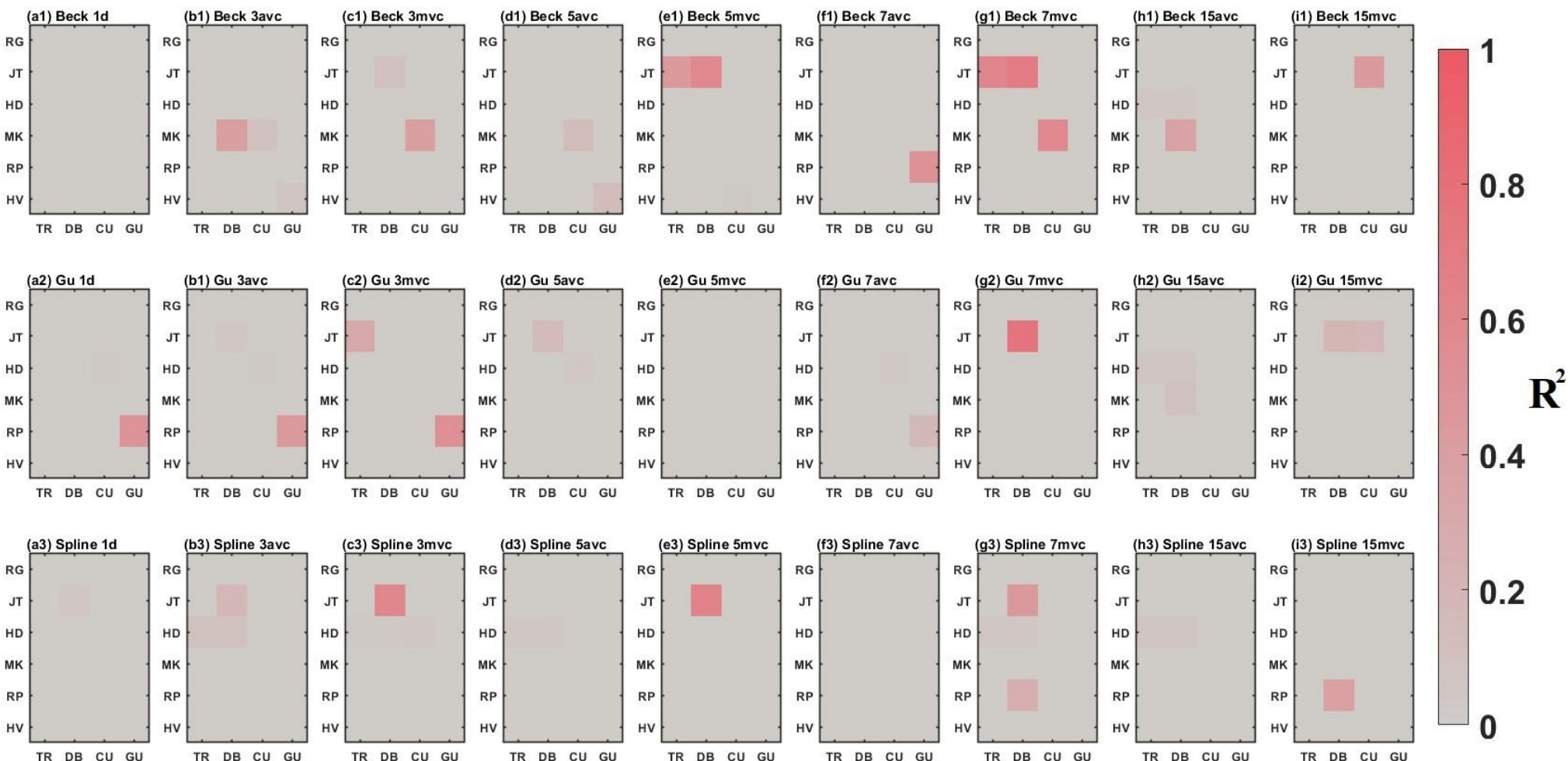

**Figure A9.** $R^2$ for each winter wheat growth stage estimations, in 2019, 2021 and 2022, using in situ SIF measured from 'morning' with varying estimation portfolios. For each growth stage estimation, the greyer sector presents the lower $R^2$ for the estimation, while the redder sector presents the higher $R^2$.

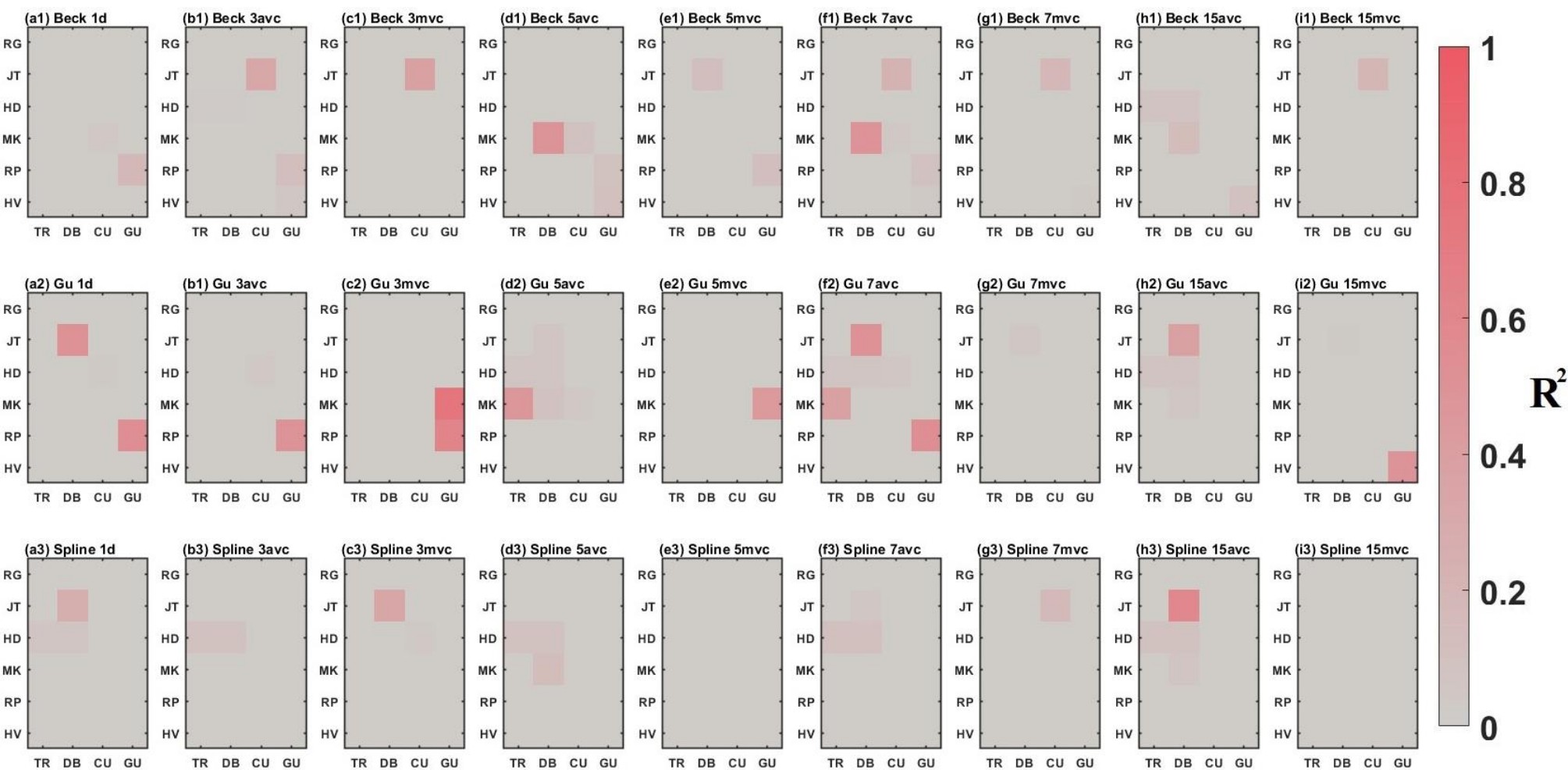

**Figure A10.** $R^2$ for each winter wheat growth stage estimations, in 2019, 2021 and 2022, using in situ SIF measured from 'afternoon' with varying estimation portfolios. For each growth stage estimation, the greyer sector presents the lower $R^2$ for the estimation, while the redder sector presents the higher $R^2$.

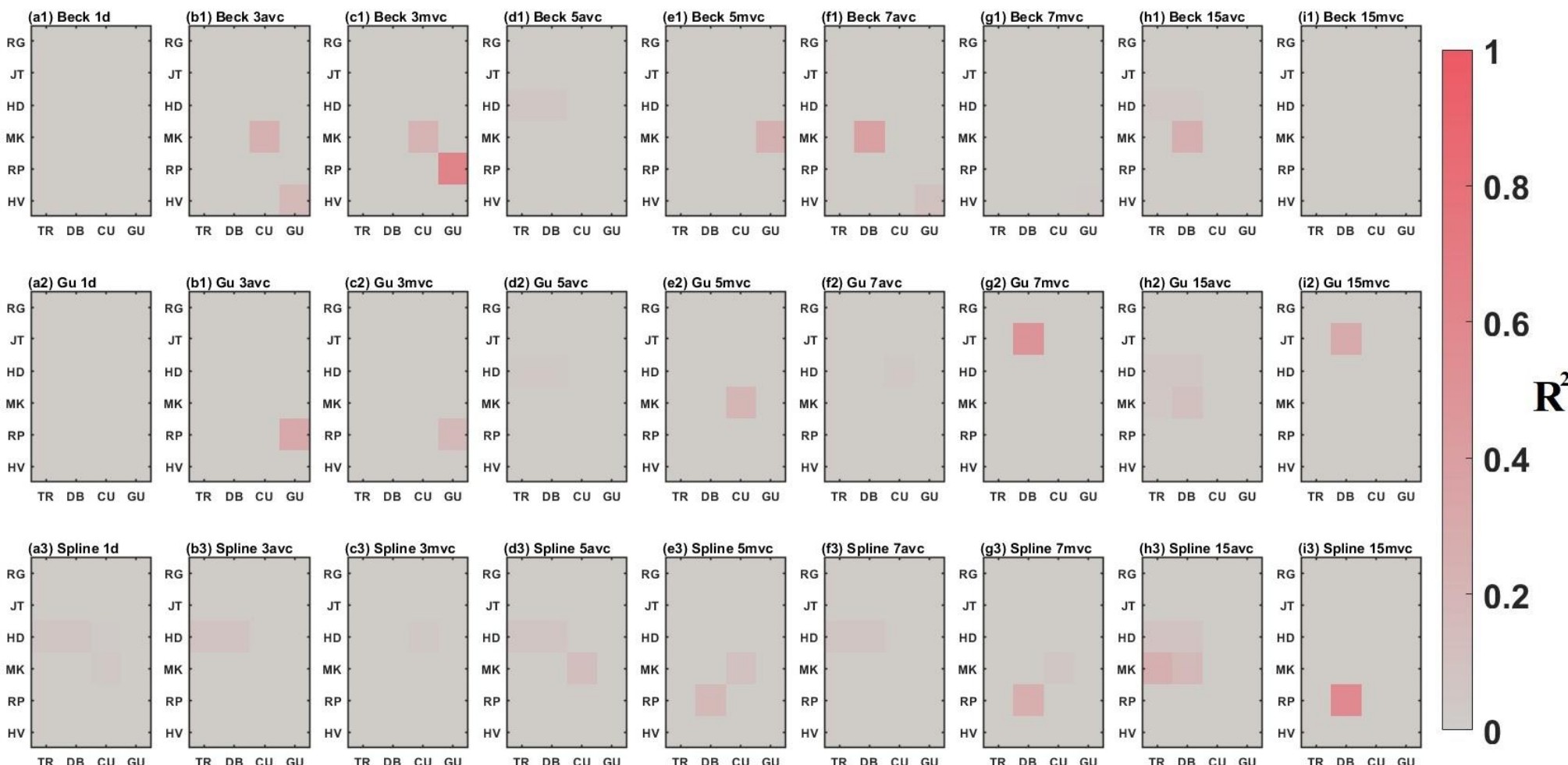

**Figure A11.** $R^2$ for each winter wheat growth stage estimations, in 2019, 2021 and 2022, using in situ SIF measured from 'afternoon' with varying estimation portfolios. For each growth stage estimation, the greyer sector presents the lower $R^2$ for the estimation, while the redder sector presents the higher $R^2$.

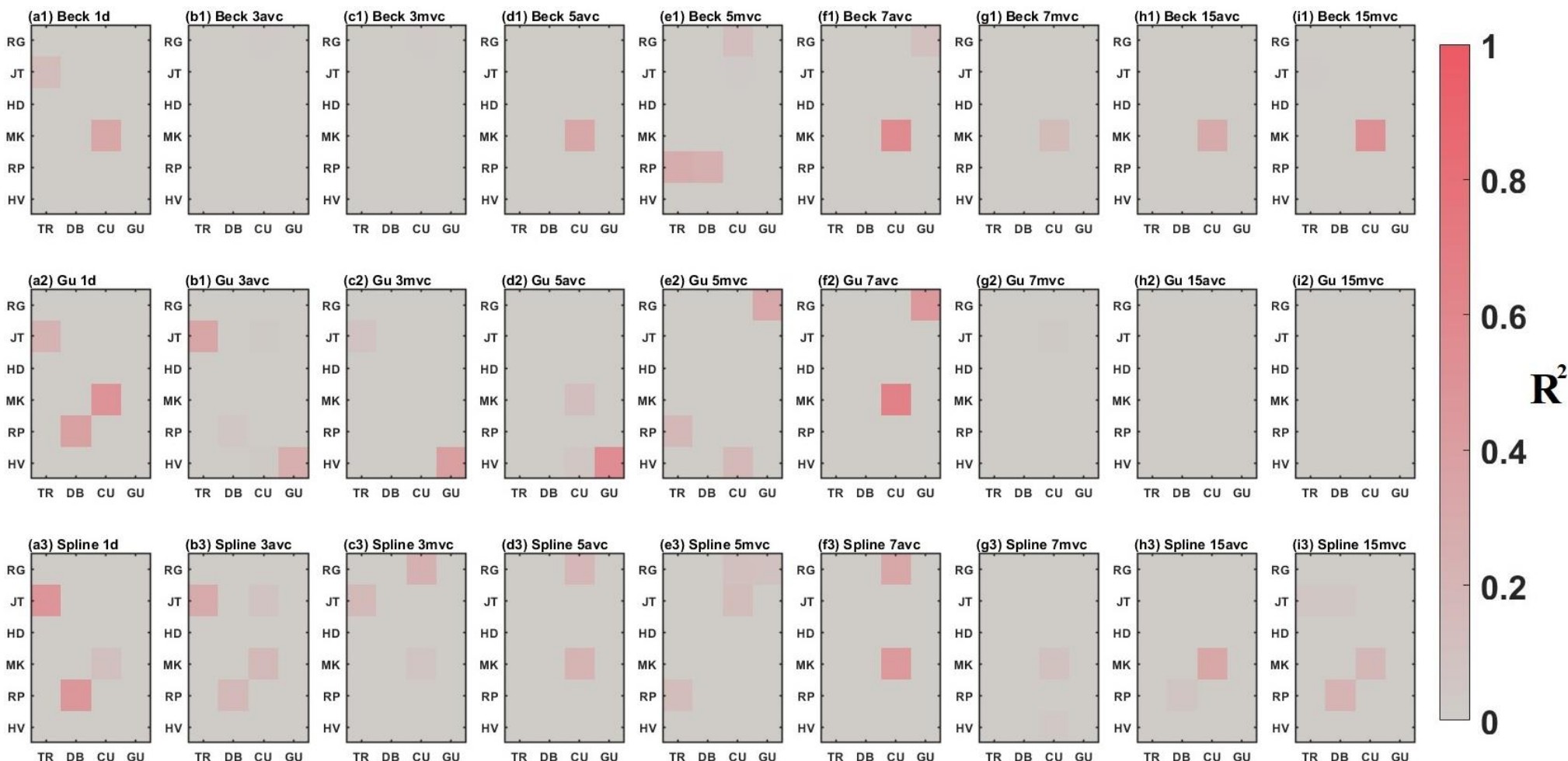

**Figure A12.** $R^2$ for each winter wheat growth stage estimations, from 2018 to 2022, using TROPOMI SIF measured from 'morning' with varying estimation portfolios. For each growth stage estimation, the greyer sector presents the lower $R^2$ for the estimation, while the redder sector presents the higher $R^2$.

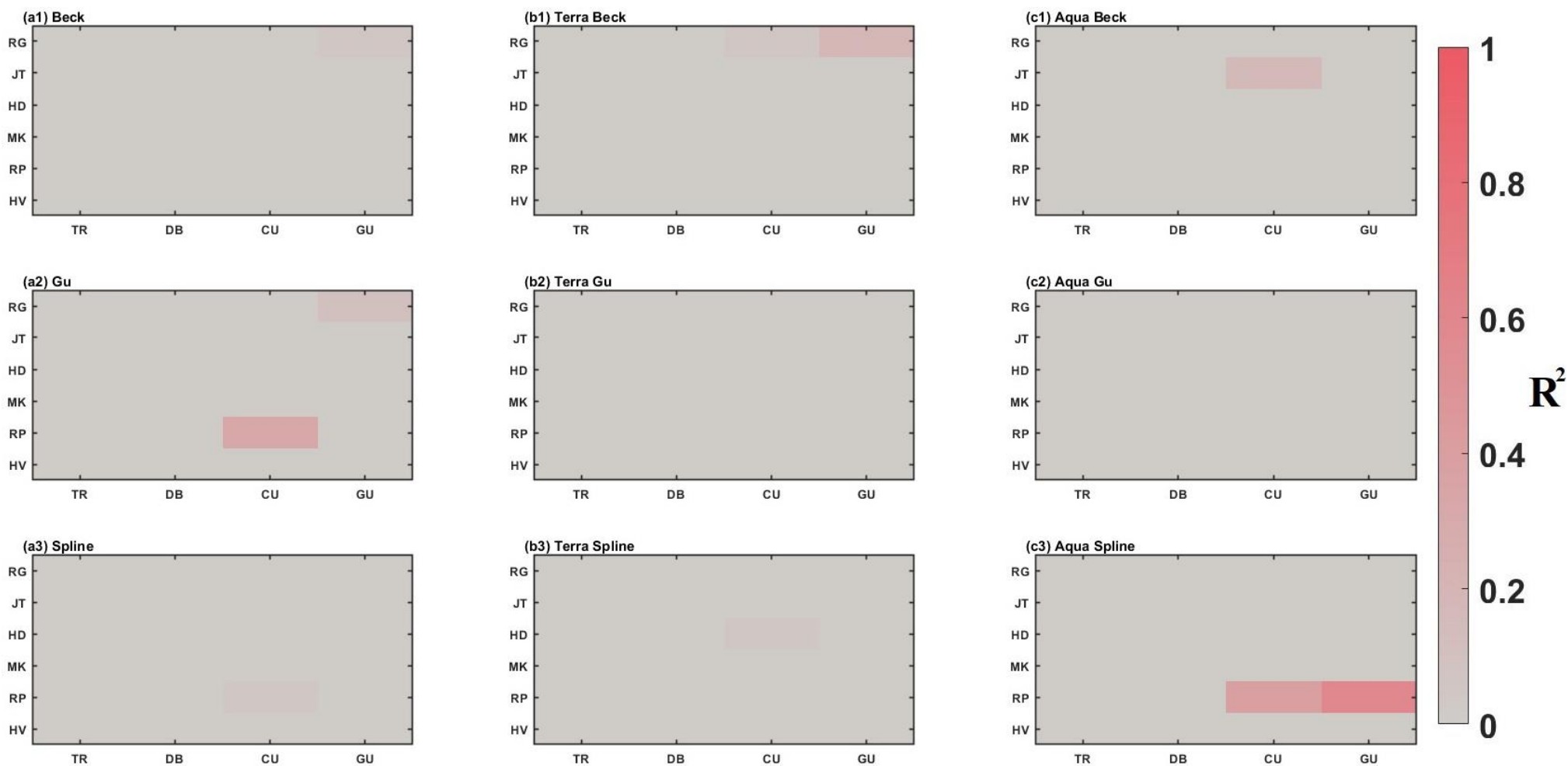

**Figure A13.** $R^2$ for each winter wheat growth stage estimations, from 2018 to 2022, using MODIS EVI measured from 'afternoon' with varying estimation portfolios. For each growth stage estimation, the greyer sector presents the lower $R^2$ for the estimation, while the redder sector presents the higher $R^2$.

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
