# Peer review of "Identifying Crop Growth Stages from Solar-Induced Chlorophyll Fluorescence Data in Maize and Winter Wheat from Ground and Satellite Measurements"

_remotesensing, doi:10.3390/rs15245689_

Round 1

Reviewer 1 Report

Comments and Suggestions for Authors

The manuscript is structured well and generally well written. Materials and methods are clearly presented. However, it would be helpful to incorporate a study area map. It is also not clear whether the locations of ground measurements were taken and how ground measurements were linked to the satellite data. Accuracy assessment should be presented under Materials and Methods section before being presented in the Results section.  Results are well presented and placed in context of existing theory. While the manuscript reads very well, it requires minor editing.  There are several cases of inadequate language use. The investigation presents an innovative way of monitoring crop growth using solar-induced chlorophyll fluorescence data based on remote sensing.

The following language d editorial issues were detected:

Line 221: this stud's - should be this study's ...........

Line 264: The use of 'It's' should be reconsidered. There are several other instances where this is used. 

Line 361: two growing season of winter wheat .... two growing seasons of winter wheat

Line 392:  date set --------dataset

Lines 741 - 745: This paragraph presents conclusions, and thus should be moved to the relevant section (5 Conclusions). Or should just be presented as a summary of study findings as was done in Lines 790 - 797.

Comments on the Quality of English Language

The use of English language is very good. Minor edits by authors should be done as indicated on the comments to them.

Author Response

Dear Reviewer 1,

Thank you very much for your time to review our manuscript. We truly appreciate your positive comments and suggestions on this work.

We have uploaded a point-to-point reply to your comments into the attachment, in which the reply is in light blue and your comments in black.

Thank you again for your time and suggestions.

Best regards,

Yuqing Hou

Reviewer 2 Report

Comments and Suggestions for Authors This paper evaluated the capacity of SIF data, obtained from both ground-based and satellite measurements, to identify crop growth stages and compared with EVI data. The study emphasised the importance of reconciling the phenological characteristics with crop growth stages and systematically evaluated a framework for crop growth stage identification with remote sensing data. The effort has been made by this study to find a set of fixed processing methods suitable for universal application, which paved way for further researches. This is a well-written paper containing interesting results which merit publication. For the benefit of the reader, however, a few points need clarifying.

Comments on the Quality of English Language

Minor editing of English language required

Author Response

Dear Reviewer 2,

Thanks for your time to review our manuscript and providing the constructive suggestions and comments!

We have uploaded a point-to-point reply to your comments into the attachment, in which the reply is in light blue and your comments in black. 

Thank you again for your time and suggestions.

Best regards,

Yuqing Hou

Reviewer 3 Report

Comments and Suggestions for Authors

This study  concerns identifying crop growth stages from solar-induced chlorophyll  fluorescence data in maize and winter wheat from ground and  satellite measurements. The investigation  shows the effectiveness of SIF data within the phenological  estimation framework for discerning the growth stages of both maize and winter wheat. The manuscript is clear and relevant int field  Remote Sensing for Precision Farminng and Crop Phenology. It's structure  is good. The experiments are appropiate  and well conducted. Conclusions are  consistent with the evidence and arguments presented.

My additional comments:

-In Eg. 1  parameters are not explained.

Author Response

Dear Reviewer 3,

Thanks a lot for your time reviewing our manuscript and your positive comment!

We have uploaded a point-to-point reply to your comments into the attachment, in which the reply is in light blue and your comments in black. 

Thank you again for your time and suggestions.

Best regards,

Yuqing Hou

Reviewer 4 Report

Comments and Suggestions for Authors

This paper gives a very thorough investigation into using SIF to enhance the identification of some major phenological events in maize and winter wheat crops using remote sensing.

L89 should be “heading” or “flowering” for wheat not “tasselling”.

Figure 1, shows quite a pronounced depression at about day 115 of the winter wheat SIF which is not reflected in the smoothed line (is this a consequence of weather conditions or changes in the reflectance spectrum of the crop around flowering).

L504 substitute “heading” for “tasseling” then in following line insert “tasseling” before “in maize”.

L515 ‘satisfactory accuracy” according to what standard? I suggest that users should be consulted about this.

L 787 ff. I was reminded when reading these sentences of when I read objective 4 mentioning modelling, I did think that there might be a comparison with, or addition of, one of many crop models to improved predictability and generality.

The reference display needs tidying and consistency.  Some references use journal full titles others just the abbreviated title.

L911. Reference 10 has no journal name

L917 similarly reference 13 which duplicates the title.

L 1064 Ref 70 either no publisher or no journal title.

Author Response

Dear Reviewer 4,

Thanks a lot for your time reviewing our manuscript and providing the constructive suggestions!

We have uploaded a point-to-point reply to your comments into the attachment, in which the reply is in light blue and your comments in black. 

Thank you again for your time and suggestions.

Best regards,

Yuqing Hou

Reviewer 5 Report

Comments and Suggestions for Authors

It should be hot topic research area to identify crop growth stage using SFI with classical NDVI or EVI and it is nice research. However, I have several comments/ questions as follow.

1)    In chapter 2.5, you set up “11 day moving window”. But, since the data are SIF daily and EVI every 8 days, why 11 days do you select? Can you explain?

2)    After formulation (3), you mention several parameters are selected as empirical parameters. How do you select those parameters?

3)    In chapter 2.5.3,  you mention there are very big challenges or limitations of your study about observation data difference between SIF and EVI. How do you justify your study without any harmonized data of SIF and EVI?

4)    In Figure 4, I have a difficult time to understand EVI. Why do you only use EVI during a certain period? Like as (b), there are only joining, regreen and harvest for EVI.

5)    In figure 6, there are big differences between your results and Area-OB for Maize.

6)    In figure 8, do you have any detail explanation about the difference between in-situ and EO for SIF and EVI. There are big differences of trend and relationship between them.

7)    In table 5 and other, you mention about the timing “morning or afternoon”. But, you didn’t explain and justify why you choose.

Author Response

Dear Reviewer 5,

Thanks for your time to review our manuscript and providing the following constructive suggestions and comments!

We have uploaded a point-to-point reply to your comments into the attachment, in which the reply is in light blue and your comments in black. 

Thank you again for your time and suggestions.

Best regards,

Yuqing Hou

Round 2

Reviewer 5 Report

Comments and Suggestions for Authors

No further comments